# Calreticulin del52 and ins5 knock-in mice recapitulate different myeloproliferative phenotypes observed in patients with MPN

Camélia Benlabiod[1,2,3], Maira da Costa Cacemiro[1,2,3,4], Audrey Nédélec[5,6], Valérie Edmond[1,2,3], Delphine Muller[1,2,3], Philippe Rameau[7], Laure Touchard[8], Patrick Gonin [8], Stefan N. Constantinescu [5,6], Hana Raslova[1,2,3], Jean-Luc Villeval[1,2,3], William Vainchenker [1,2,3], Isabelle Plo[1,2,3,9] & Caroline Marty [1,2,3,9 ✉]

Somatic mutations in the calreticulin (*CALR*) gene are associated with approximately 30% of essential thrombocythemia (ET) and primary myelofibrosis (PMF). *CALR* mutations, including the two most frequent 52 bp deletion (*del52*) and 5 bp insertion (*ins5*), induce a frameshift to the same alternative reading frame generating new C-terminal tails. In patients, *del52* and *ins5* induce two phenotypically distinct myeloproliferative neoplasms (MPNs). They are equally found in ET, but *del52* is more frequent in PMF. We generated heterozygous and homozygous conditional inducible knock-in (KI) mice expressing a chimeric murine CALR del52 or ins5 with the human mutated C-terminal tail to investigate their pathogenic effects on hematopoiesis. Del52 induces greater phenotypic changes than ins5 including thrombocytosis, leukocytosis, splenomegaly, bone marrow hypocellularity, megakaryocytic lineage amplification, expansion and competitive advantage of the hematopoietic stem cell compartment. Homozygosity amplifies these features, suggesting a distinct contribution of homozygous clones to human MPNs. Moreover, homozygous *del52* KI mice display features of a penetrant myelofibrosis-like disorder with extramedullary hematopoiesis linked to splenomegaly, megakaryocyte hyperplasia and the presence of reticulin fibers. Overall, modeling *del52* and *ins5* mutations in mice successfully recapitulates the differences in phenotypes observed in patients.

---

[1] INSERM, UMR 1287, Gustave Roussy, Villejuif, France. [2] Université Paris-Saclay, UMR 1287, Gustave Roussy, Villejuif, France. [3] Gustave Roussy, UMR 1287, Villejuif, France. [4] Department of Clinical Analysis, Toxicology and Food Sciences, School of Pharmaceutical Sciences of Ribeirão Preto, University of São Paulo-USP, Ribeirão Preto, São Paulo, Brazil. [5] Ludwig Institute for Cancer Research, Brussels, Belgium. [6] de Duve Institute, Walloon Excellence in Life Sciences and Biotechnology (WELBIO), Université catholique de Louvain, Brussels, Belgium. [7] Integrated Biology Core Facility, Gustave Roussy, Villejuif, France. [8] Preclinical Research Plateform, Unité Mixte de Service AMMICA 3655/US 23, Gustave Roussy, Villejuif, France. [9] These authors contributed equally: Isabelle Plo, Caroline Marty. ✉email: caroline.marty@gustaveroussy.fr

Myeloproliferative neoplasms (MPNs) are driven by the acquisition of somatic mutations in the hematopoietic stem cell (HSC) compartment leading to the constitutive activation of the JAK-STAT signaling pathway. Essential thrombocythemia (ET) and primary myelofibrosis (PMF) are two disorders of the megakaryocytic lineage defined by an increased level of circulating platelets in ET and by megakaryocytic hyperplasia/dysplasia accompanied by the development of fibrosis in the bone marrow (BM) in PMF. PMF can display several other clinical features, including leukocytosis, splenomegaly, due to extramedullary hematopoiesis and anemia that may progress into pancytopenia. ET can evolve into a pre-fibrotic stage and overt secondary myelofibrosis (MF). The main driver mutations of ET and PMF are $JAK2^{V617F}$, mutations in the exon 9 of the calreticulin (CALR) gene, and in the thrombopoietin (TPO) receptor MPL gene[1–4]. Several CALR mutations have been reported and all induce a +1-base pair (bp) frameshift to an alternative reading frame leading to the replacement of the negatively-charged C-terminal tail to a new positively charged sequence with 34 residues common to all mutants. It was shown that CALR mutants specifically bind to and constitutively activate MPL, explaining the megakaryocyte (MK) phenotype of the CALR-mutated MPNs[5,6].

The two most frequent CALR mutations are a 52-bp deletion (del52) or type 1 and a 5-bp insertion (ins5) or type 2. The frequency of the two types of mutations is close in ET while del52 and rarer type 1-like mutations are largely predominant in MF, thus, del52 and ins5 define two phenotypically distinct MPNs in human[7]. Retroviral (RV) modeling in mice has shown that the two types of human CALR mutations induce a MPL-dependent phenotype mimicking human disorders, but this approach presents pitfalls such as the ectopic over-expression of CALR mutants compared to endogenous wild-type (wt) CALR[8,9]. Several other models have been published including a human del52 transgenic (TG) mouse[10], CRISPR-Cas9-mediated both murine del19 and murine del52[11], and murine del61 knock-in (KI) mice[12] as well as an inducible conditional KI expressing murine CALR with the human del52-mutated end tail[13]. Yet, none of these studies directly compare the effects of in vivo physiopathological expression of the two most frequent types of CALR mutations on hematopoiesis, nor do they specifically address the disease outcomes of ins5 or type 2-like mutant expression in mice.

Understanding the basis of the phenotypic differences induced by type 1 and type 2 mutations is of deep interest as their impact on the evolution of the disease and thus patient prognosis differ. In the present work, we conducted a direct comparison of the effects of heterozygous and homozygous physiopathological expression of del52 and ins5 mutants on hematopoiesis using inducible conditional KI mice. We chose to express chimeric mouse/human proteins based on the fact that mutated C-terminal tails of the two species display different amino acid composition and we thus generated these del52 and ins5 KI mice before the recent demonstration that the human and murine mutated C-terminal tails have close oncogenic properties[11,14]. Expression of del52 induces a stronger phenotype including thrombocytosis, leukocytosis, megakaryocytic and progenitor amplification as well as a penetrant fibrosis in spleen compared to ins5. Moreover, these phenotypes are both amplified by homozygosity suggesting a possible distinct contribution of the homozygous clone to the development and the extent of the human disorders. Thus, our KI mice recapitulate the human MPNs and provide powerful preclinical models to decipher the molecular bases of the differences between del52 and ins5 diseases.

## Results

### Generation of KI mice for CALR del52 and CALR ins5. We used the FLEx strategy to generate conditional heterozygous mice

carrying a floxed construct fusion of mouse Calr exons 8 and 9 followed with wild-type mouse Calr exon 8 and a modified exon 9 encompassing either the murine deletion of 52-bp (CALR del52) or a 5-bp insertion (CALR ins5) analogous to human mutations and the human mutated C-terminal sequence. To generate KI mice, floxed mice were crossed with SCL-CreER$^T$ TG animals expressing a tamoxifen-inducible Cre. Thus, after induction of the KI allele expression by Cre recombinase these mice express heterozygous mouse CALR with human mutated C-terminal tail from its endogenous promoter (Fig. 1a, b). We purified BM lineage-negative (Lin$^-$) cell population for each KI model and measured CALR protein levels by Western blot (Fig. 1c). Using an antibody (Ab) (CALR-tot) that recognizes both CALR wt and mutants we mostly detected doublet bands at the expected CALR protein size that might reflect the presence of a cleaved product and/or protein post-translational modifications. Identical result was obtained using a different Ab clone (Supplementary Fig. 1a) and expression levels of three independent experiments were quantified (Supplementary Fig. 1b). We observed no difference in CALR expression between wt (+/+) and heterozygous +/del52 and +/ins5 cells. However, we noticed that in the absence of endogenous CALR wt (homozygous del52/del52 and ins5/ins5 cells) the amount of CALR mutant protein in cell lysate was lower (only significant with 1.8−fold less CALR del52 detected in del52/del52 compared to +/+ cells) than in both wt and heterozygous cells. Using an Ab specifically directed against the mutated C-terminal tail (CALR-Cter) that does not detect endogenous CALR wt, there seems to be less CALR del52 than CALR ins5 in the homozygous cells, in accordance with the difference of protein levels revealed with the CALR-tot Ab. Surprisingly, we barely observed any band in the heterozygous cell lysates.

### KI mouse phenotypes depend on the mutation and zygosity.
Both del52 and ins5 KI mice developed a rapid thrombocytosis after induction by tamoxifen compared to wt (+/+) mice, but at higher levels in the presence of del52 than in the presence of ins5 (Fig. 2a, b, Supplementary Fig. 1c). Moreover, we observed that the thrombocytosis was more elevated in homozygous compared to heterozygous context. Both del52 and ins5 KI mice, especially in the homozygous context, also developed a significant leukocytosis (Fig. 2c, d). Compared to wt controls, hemoglobin levels were normal in all KI models except significantly lower in homozygous del52 mice and starting to decrease in 7-month-old homozygous ins5 mice (Fig. 2e, f).

Leukocytosis developed by KI mice, especially in the homozygous setting, was unexpected from CALR-mutated patient studies and was due to a general increase in the different white blood cell (WBC) populations (Supplementary Fig. 2a, b). There was a significant increase in neutrophils, lymphoid cells, and eosinophils. Monocytosis was higher in presence of del52 than in the presence of ins5, while basophilia was more important in presence of ins5 than del52 (Supplementary Fig. 2a, b). The del52 mutation was shown to lead to a general increase of various T lymphocyte sub-populations (Supplementary Fig. 3a, b).

### Effect of the CALR mutations on the megakaryocytic lineage.
We studied the effect of del52 and ins5 on the MK lineage, and observed a TPO-independent MK progenitor (CFU-MK) growth in heterozygous and homozygous del52 and ins5 KI mice, as stimulation by TPO induced only a significant increase of CFU-MK frequency in +/+ mice (Fig. 3a). Moreover, we measured the frequencies of MK progenitors (MkPs) and MKs (CD41$^+$CD42$^+$) in BM (Fig. 3b) and spleen (Fig. 3c) of KI mice by flow cytometry. We observed increased MkPs and MKs in del52/del52 KI mice and to a lower extent for MkPs in ins5/ins5 KI mice. Accordingly,

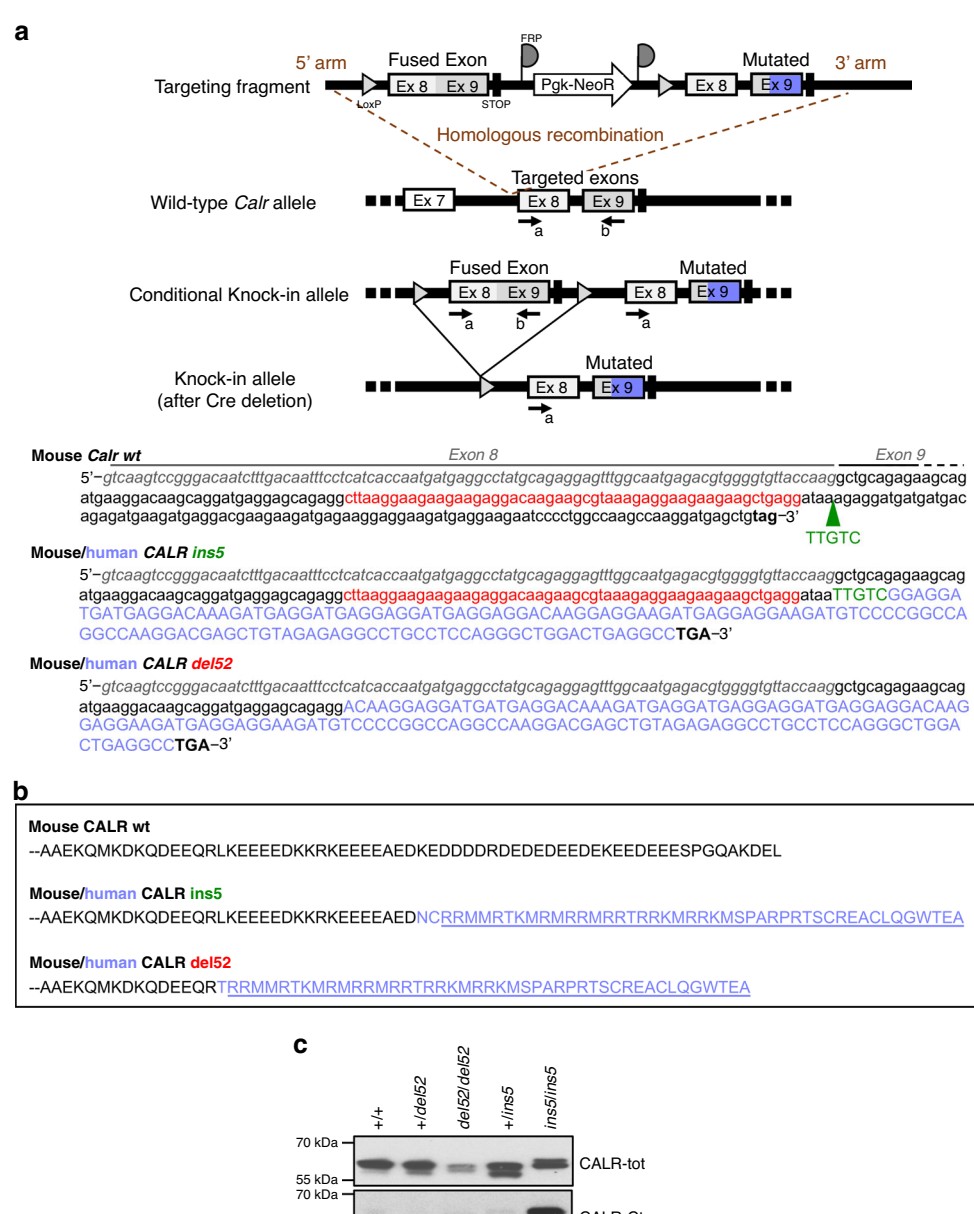

**Fig. 1 Conditional *CALR del52* and *CALR ins5* KI mouse models. a** Schematic representation of the wt and conditional KI alleles obtained after crossing the chimeras with flippase TG mice. KI allele was obtained after crossing mice with SCL-CreER[T] TG mice and induction of Cre-mediated deletion of the wt fused exon 8 and 9 by tamoxifen, arrows are LoxP sites. The heterozygous and homozygous status of conditional KI mice was verified by DNA genotyping using a and b primers. A similar strategy was used to genotype conditional *ins5* and *del52* KI mice. Sequence of mouse *Calr* (lower cases) exon (Ex) 8 (italic gray) and Ex 9 (black) was depicted with the 52-bp (red) deleted in the mouse/human *CALR del52* chimera and the green arrow pointing to the position of the 5-bp insertion (TTGTC in green) in *CALR ins5*. Human sequence is depicted in upper cases and the purple sequence encodes the human mutated C-terminal tail. **b** Protein sequences of mouse CALR wt Ex 9, and CALR del52 and CALR ins5 mutated Ex 9. In black is mouse sequence and in purple human sequence with the common mutated C-terminus underlined. **c** Lin⁻ cells isolated from BM of wt (+/+), heterozygous (+/*del52* or +/*ins5*) and homozygous (*del52/del52* and *ins5/ins5*) *CALR*-mutated KI mice were analyzed by Western blot for expression of CALR using an Ab directed against the N-terminus, common to both wt and mutant CALR (CALR-tot, Cell Signaling) and an Ab specific of the mutated C-terminal tail (CALR-Cter). ß-Actin was used as loading control. Blots shown are representative of at least three independent experiments. Source data are provided as a Source data file.

immuno-histological staining for von Willebrand factor (vWF) confirmed MK hyperplasia in BM (Fig. 3d) and spleen (Supplementary Fig. 4a) of homozygous *del52* mice while homozygous *ins5* mice mostly presented an apparent increase in the MK size and the number of MKs but only in the spleen. Milder phenotype was found with heterozygous *del52*. Independently of zygosity, all KI mice except +/*ins5* animals, displayed giant MKs with

polylobulated nuclei suggesting increased modal ploidy. Indeed, in vivo MKs achieved a significantly higher zygosity-dependent ploidy in *del52* KI mice than in wt littermates (Fig. 3e). The mean ploidy of *ins5/ins5* MKs was identical to *del52/del52* MKs, both displaying also a modal ploidy of 32N compared to 16N in wt littermates (Supplementary Fig. 4b), while there was no marked difference between ploidy of +/*ins5* and wt MKs with the same

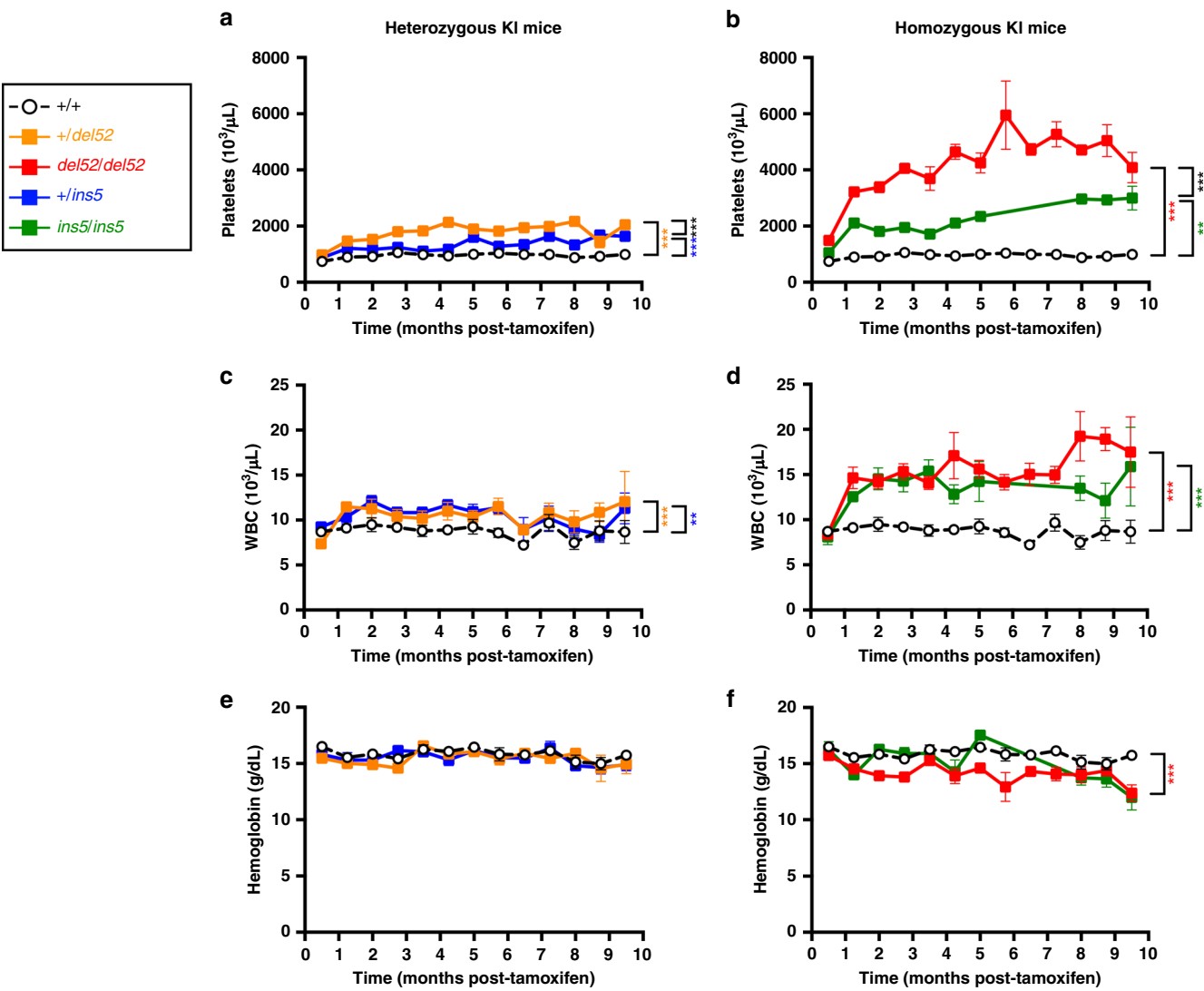

**Fig. 2 Severity of KI mouse phenotype depends on the mutation and the zygosity.** Blood parameters including **a**, **b** platelets, **c**, **d** WBC counts and **e**, **f** hemoglobin levels were determined in wt $+/+$ (open circles with connecting black dashed line, $n = 16, 18, 15, 11, 11, 8, 13, 7, 14, 11, 5, 6, 4$ for platelets, $n = 34, 25, 26, 25, 19, 18, 17, 12, 19, 13, 10, 13, 7$ for WBC and $n = 36, 26, 28, 25, 21, 19, 20, 11, 19, 11, 9, 14, 9$ for hemoglobin), heterozygous $+/del52$ (orange squares and solid line, $n = 18, 20, 15, 12, 10, 7, 11, 7, 12, 12, 8, 5, 10$ for platelets, $n = 18, 19, 15, 13, 11, 7, 11, 7, 14, 11, 7, 5, 3$ for WBC and $n = 19, 20, 16, 13, 11, 8, 11, 7, 13, 9, 7, 5, 3$ for hemoglobin) and $+/ins5$ (blue squares and solid line, $n = 20, 7, 13, 18, 11, 13, 10, 5, 4, 4, 5, 8, 7$ for platelets, $n = 21, 8, 12, 19, 11, 12, 10, 5, 5, 4, 5, 8, 9$ for WBC and $n = 20, 7, 12, 18, 11, 13, 13, 5, 5, 4, 5, 8, 7$ for hemoglobin) mice after tamoxifen induction. $N$ represents the number of individual mouse and varies depending on time of sampling during the course of the experiment, as indicated in the Source data file. Data are the means ± SEM. Significance was determined using ANOVA with Tukey's multiple comparison test: $**p = 0.001$, $***p \leq 0.0001$ (**a**, **b**); $**p \leq 0.001$, $***p = 0.0007$ (**c**); $***p < 0.0001$ (**d**); $***p = 0.0008$ (**f**). Source data are provided as a Source data file.

modal ploidy. However, in comparison to wt MKs, the 8N peak decreased in favor of the 32N peak in $+/ins5$ MKs. These data suggest that $del52$ induces a stronger amplification of MK progenitor cells compared to $ins5$ (increase in MkP and MK numbers) while both mutations affect the late stages of megakaryopoiesis (increase in MK ploidy), indicating that CALR ins5 seems less effective than CALR del52.

***CALR del52* KI mice develop a myelofibrosis-like disease.** Mice were analyzed after 10 months of tamoxifen induction. There was a significant decrease in BM cellularity of $del52/del52$ mice combined with an increase in spleen weight (Fig. 4a, b). There was barely a difference in the $+/del52$, $ins5/ins5$, and $+/ins5$ mice compared to

wt controls. In the $del52/del52$ mice, the 4-fold increase in the frequency of giant MKs in the BM (Fig. 3) may partially explain the decrease in BM cellularity. Moreover, in the BM of $del52/del52$ mice, we observed a decrease in the frequencies of both erythroid progenitors (BFU-E) and erythroblasts (Ter-119[+]) (Supplementary Fig. 5a) with no noticeable change in granulo-monocytic progenitors (CFU-GM) and granulocyte precursors (Mac1[+]Gr1[+]) (Supplementary Fig. 5c) or the lymphoid cells (CD3[+] and B220[+]) (Supplementary Fig. 5e). Histological examination by silver staining showed a very mild MF in $del52/del52$ mice but no significant development of reticulin fiber network in the BM of any other KI mice (Fig. 4c, BM, reticulin silver (RET) staining), but we observed a neo-osteogenesis only in rare cases of $del52/del52$ mice.

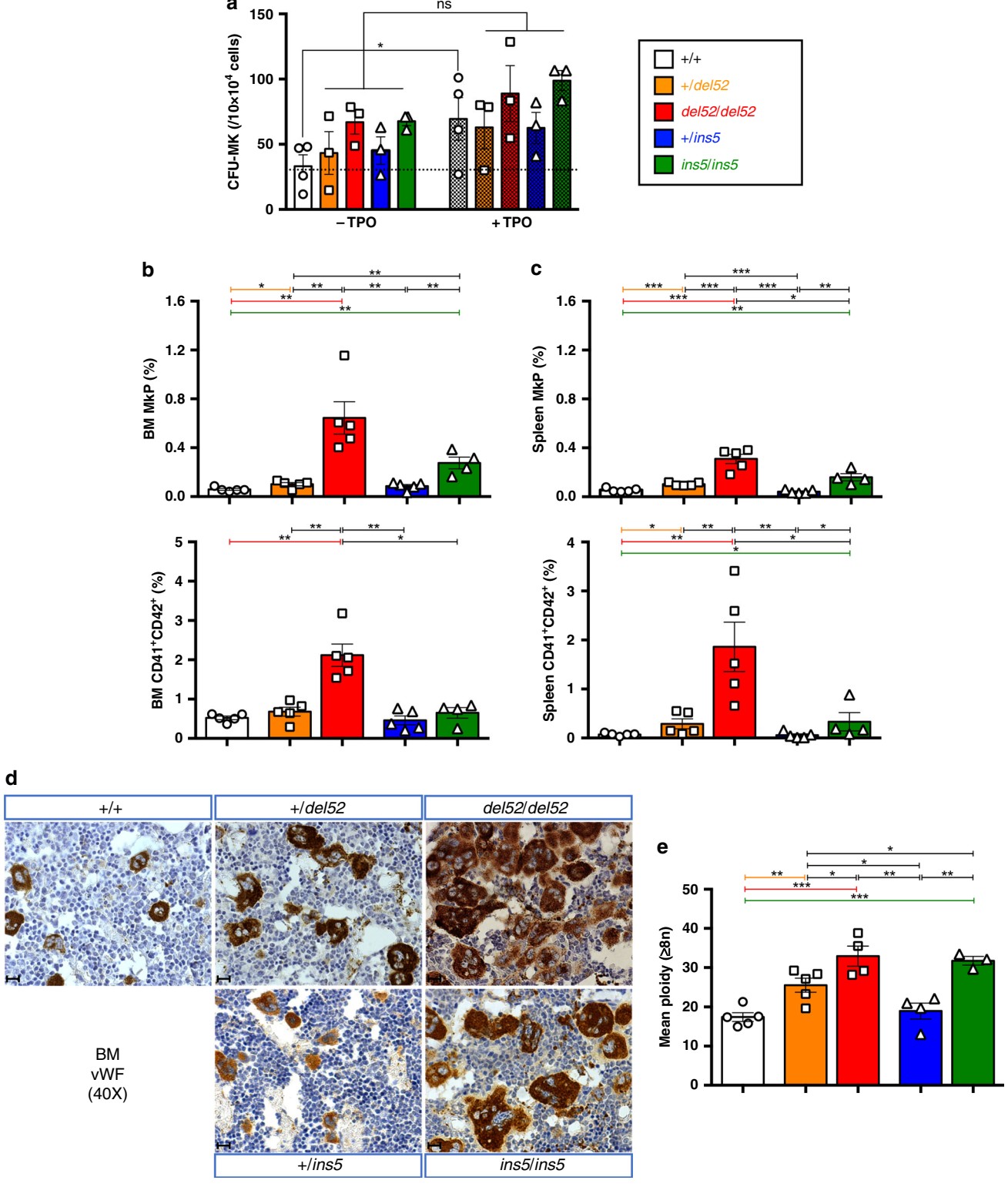

The unexpected quasi-absence of fibrosis in BM, particularly of homozygous *del52* KI mice that present an important MK hyperplasia confirmed on hematoxylin/eosin/safran (HES) staining (Fig. 4c), prompted us to explore fibrosis in spleen. Histology revealed the presence of a thicker reticulin fiber network in homozygous *del52* compared to wt spleen, and in a less penetrant fashion in homozygous *ins5* KI mice (Fig. 4c, Spleen, RET). Spleen structure was partially disorganized in the +/*del52* and *ins5*/*ins5* KI mice and, in *del52*/*del52* KI mice, white pulp territories were almost lost (Supplementary Fig. 6), attributable to a strong MK hyperplasia (Fig. 4c, Spleen, HES) and an increase in RBCs (Fig. 4d) that was confirmed by semi-solid cultures and flow cytometry analysis at the progenitor and erythroblast levels (Supplementary Fig. 5b). An increase in granulo-monocytic lineage cells was also observed only in *del52*/*del52* KI spleens (Supplementary Fig. 5d), explaining the higher spleen weights of these mice compared to the three other phenotypes and the wt littermates (Fig. 4b). Disorganization of

**Fig. 3 Amplification of the megakaryocytic lineage in KI mice. a** Frequencies of BM megakaryocytic progenitors (CFU-MKs) with or without TPO of ≥10 months post-tamoxifen KI mice were represented as means ± SEM ($n = 4$ individual +/+ mice and $n = 3$ mice for heterozygous and homozygous genotypes). In the figure, white is used for +/+, orange for +/del52, red for del52/del52, blue for +/ins5 and green for ins5/ins5 genotypes. Significance of +TPO (hatched histograms) compared to the −TPO (solid histograms) condition was assessed using a two-sided parametric paired $t$-test (*$p = 0.0192$; ns not significant). **b, c** Frequencies of megakaryocytic progenitors (MkPs, Lin⁻c-Kit⁺Sca-⁻CD150⁺CD41⁺) and MKs (CD41⁺CD42⁺) in **b** BM and **c** spleen cells from +/+, +/del52, and del52/del52 compared to +/ins5 and ins5/ins5 mice ≥10 months after tamoxifen induction were expressed as means ± SEM ($n = 5$ individual mice except $n = 4$ ins5/ins5 mice). Significance for MkPs was calculated with a two-sided parametric $t$-test and for MKs with a two-sided nonparametric Mann–Whitney test. In **b**, *$p = 0.023$; **$p = 0.0023$ (+/+ vs. del52/del52), 0.0016 (+/+ vs. ins5/ins5), 0.0036 (+/del52 vs. del52/del52), 0.0063 (+/del52 vs. ins5/ins5), 0.0031 (del52/del52 vs. +/ins5), 0.004 (+/ins5 vs. ins5/ins5) in MkP analysis and *$p = 0.0159$; **$p = 0.0079$ in CD41⁺CD42⁺ analysis. In **c**, *$p = 0.0213$ (del52/del52 vs. ins5/ins5); **$p = 0.0058$ (+/+ vs. ins5/ins5), 0.003 (+/ins5 vs. ins5/ins5); ***$p = 0.0007$ (+/+ vs. +/del52), 0.0002 (+/+ vs. del52/del52), 0.0008 (+/del52 vs. del52/del52), <0.0001 (+/del52 vs. +/ins5), 0.0001 (del52/del52 vs. +/ins5) in MkP analysis and *$p = 0.0159$ (+/+ vs. +/del52), 0.0238 (+/+ vs. ins5/ins5), 0.0317 (del52/del52 or +/ins5 vs. ins5/ins5); **$p = 0.0079$ in CD41⁺CD42⁺ analysis. **d** BM MKs were immunostained for vWF. Images were obtained using a DM2000 Leica microscope and a DFC300FX Leica camera with Leica Application Suite v.2.5,OR1 acquisition software (×40 magnification). Scale bars represent 50 μm. Similar results were obtained with analyzing at least three independent mice for each genotype. **e** Mean ploidy of ≥8N MKs from ≥10 months post-tamoxifen KI mice was expressed as means ± SEM ($n = 5$ individual +/+ and +/del52 mice, $n = 4$ del52/del52 and +/ins5 mice and $n = 3$ ins5/ins5 mice). Statistical significance was determined using a two-sided parametric $t$-test: *$p = 0.0453$ (+/del52 vs. del52/del52), 0.0454 (+/del52 vs. +/ins5), 0.0498 (+/del52 vs. ins5/ins5); **$p = 0.0047$ (+/+ vs. +/del52), 0.0052 (del52/del52 vs. +/ins5), 0.0042 (+/ins5 vs. ins5/ins5); ***$p = 0.0005$ (+/+ vs. del52/del52), 0.00001 (+/+ vs. ins5/ins5). Source data are provided as a Source data file.

white pulp in spleens of homozygous del52 and, to a lesser extent, ins5 animals was accompanied with a decrease in the frequencies of lymphocytes, but their absolute numbers increased non-significantly relative to splenomegaly (Supplementary Fig. 5f). These results show that del52/del52 in contrast to ins5/ins5 mice present early signs of myelofibrosis characterized by a BM cellularity decrease, splenomegaly and extramedullary hematopoiesis even if distinct reticulin fibers could not be significantly detected in the BM in contrast to the spleen.

**Phenotype severity depends on MPL/JAK2/STAT activation.** Since fibrosis was milder in BM compared to spleen, we tested whether CALR del52 might induce a weaker signaling in BM compared to spleen MKs by phosphoflow. CALR del52 induced the constitutive phosphorylation of STAT5 in homozygous BM MKs compared to wt (+/+) littermate BM MKs that was dependent on JAK2 as level returned to basal with the JAK1/2 inhibitor ruxolitinib (Supplementary Fig. 7a). Interestingly, TPO stimulation of del52/del52 MKs increased the level of phosphorylated STAT5 suggesting that at least a certain amount of MPL was accessible for TPO binding in presence of CALR del52. However, we did not observe any obvious difference between del52-induced phosphorylation of STAT5 in BM compared to spleen MKs. In our mice, the KI allele encodes the murine (mu) CALR ins5 or del52 with human (hu) mutated C-terminal tail. In order to determine the efficacy of these chimeric (chim) CALR mutants, compared to hu CALR del52 and ins5, to activate the mu MPL, we measured STAT5 activation by luciferase assay in γ2A cells. In this system, mu MPL was indistinctively and spontaneously (−TPO) activated by chim CALR del52, chim CALR ins5, hu CALR del52 and hu CALR ins5 (Fig. 5a). As previously shown for the hu CALR mutants, the chim CALR mutants were also less efficient in constitutively activating mu MPL compared to hu MPL. As expected, stimulation of mu/hu MPL with TPO abolished the differences in STAT5 activation in presence of any of the CALR mutants and CALR wt. In order to verify whether the mild MF phenotype developed by del52/del52 KI mice was due to sub-optimal stimulation of mu MPL by the chim CALR del52 protein, we treated 1 month post-induction homozygous KI mice with the MPL agonist romiplostim, that was previously reported to induce MF in wt mice[15]. The romiplostim-treated homozygous del52 KI mice developed a significant BM hypocellularity and splenomegaly compared to treated wt littermates (Fig. 5b) accompanied with an increase in

platelets, WBC, and slight anemia (Supplementary Fig. 7b) possibly indicating the development of MF. Romiplostim treatment induced MK hyperplasia revealed by vWF immunostaining and fibrosis with the presence of black reticulin fibers in both BM and spleen of wt and, to a stronger extent, of del52/del52 mice (Fig. 5c). Taken together, this suggests that the del52/del52 KI mice develop a fibrosis in both BM and spleen linked to a constitutive stimulation of mu MPL, but mild as this activation is weak.

**Amplification of the HSC compartment in KI mice.** The study of the different cell compartments suggested that del52 may provide a stronger advantage to HSC (Lin⁻Sca-1⁺c-Kit⁺CD48⁻CD150⁺, called SLAM) than ins5. In homozygous KI mice after 10 months post-induction, there was a significant amplification of HSC in both BM and spleen that was stronger in presence of del52 compared to ins5 (Fig. 6a and Supplementary Fig. 8). An increasing trend was observed in +/del52 mice while +/ins5 was very similar to controls. To decipher the effect of both mutations in the HSC compartment, we performed a competitive BM transplantation with increasing percentages of non-induced homozygous del52 or homozygous ins5 cells with wt GFP⁺ cells into lethally irradiated recipient mice (Fig. 6b). One month after transplantation KI expression was induced by tamoxifen. Del52-engrafted mice developed a higher dose-dependent thrombocytosis (Fig. 6c) and leukocytosis (Fig. 6d) than ins5-engrafted mice and no change in hemoglobin (Supplementary Fig. 9). After 4 months, thrombocytosis developed by mice engrafted with 100% and 75% of mutated cells reached similar levels than the respective age-related (4–5 months post-induction) native del52/del52 and ins5/ins5 mice (Fig. 2b). In contrast to del52-engrafted mice, there was no obvious development of a disease at 4 months with <50% of ins5-engrafted cells. Evolution of mutated cells was followed in blood platelets, granulocytes and erythrocytes. The homozygous del52 BM cells were able to outcompete wt BM cells from as low as an initial 10% competitive engraftment, reaching 100% at 4 months with a delay in the erythrocytes likely due to the longer half-life of these cells (Fig. 7a). In contrast, competition by ins5 BM cells was slower especially when <50% of mutated cells were initially engrafted (Fig. 7b) suggesting that del52 provides a stronger selective advantage than ins5 to the HSC. Especially, percentages of ins5-mutated cells engrafted at 10% and 25% followed over a 4-month period after KI induction were visibly below values for similar amounts of engrafted del52-mutated

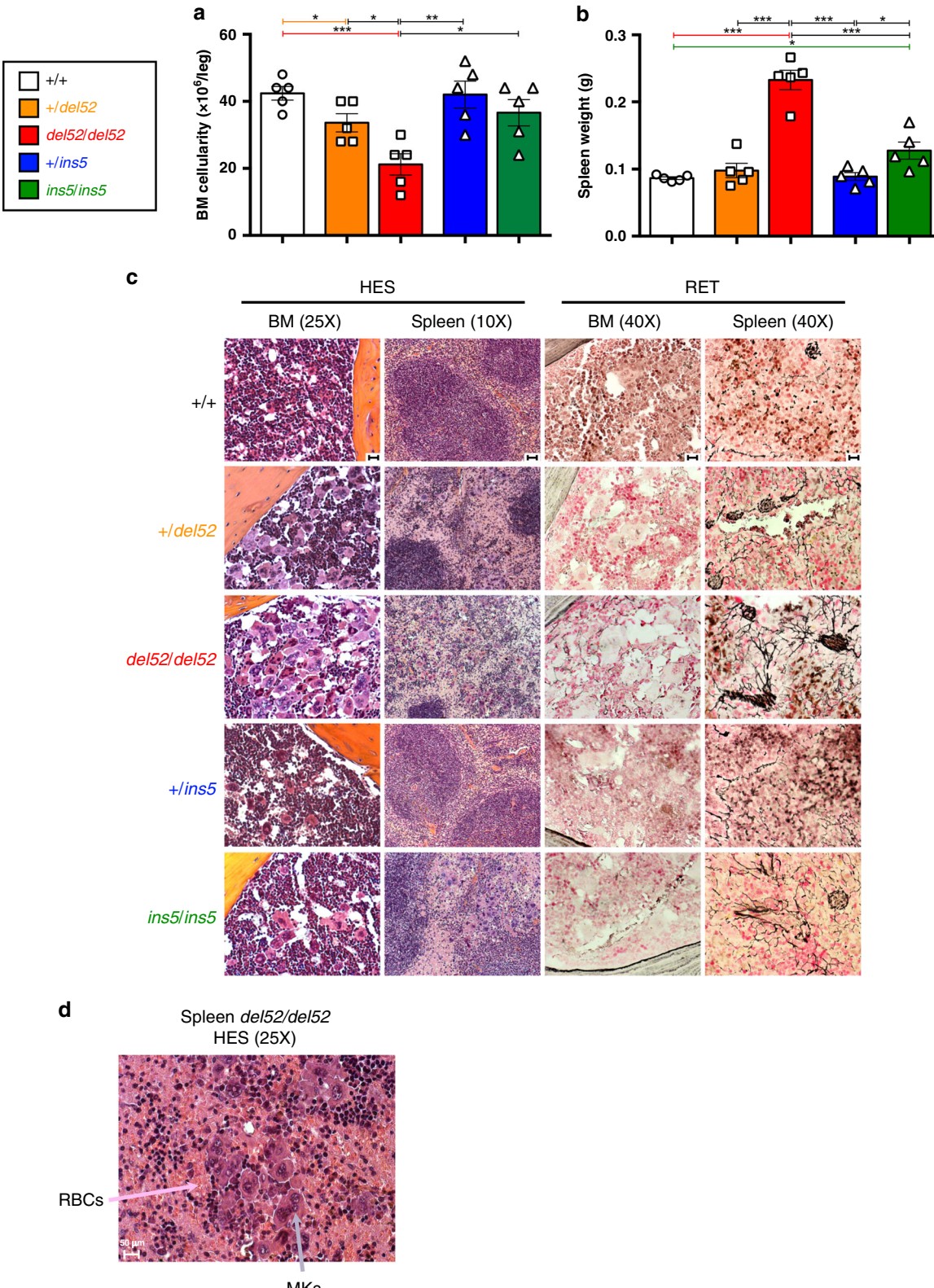

cells. Similarly, 4 months after induction of KI expression, HSC-enriched BM cell populations (Lin⁻Sca-1⁺c-Kit⁺/LSK and SLAM) were 100% *del52*-mutated independently of the initial ratio engrafted while for initial ratios of 25% and 10% *ins5* BM cells, HSC populations still contained wt cells (Fig. 7c). More interestingly, the frequencies of SLAM in BM dose-dependently increased with higher initial amounts of transplanted *del52/del52* BM cells whereas frequencies of *ins5/ins5* SLAM remained independent of the initial percentages of engrafted mutated BM cells, suggesting a *del52*-induced increase in HSC symmetric division compared to *ins5* (Fig. 7d). Taken together these results suggest that both *del52* and *ins5* provide a competitive advantage to HSC but *del52* induces a stronger amplification of the HSC compartment than *ins5*.

**Fig. 4 BM and spleen features of KI mice. a** BM cellularity and **b** Spleen weight of ≥10 months post-tamoxifen KI mice were expressed as means ± SEM (n = 5 different mice for each genotype). In the figure, white is used for +/+, orange for +/del52, red for del52/del52, blue for +/ins5 and green for ins5/ins5 genotypes. Significance was calculated with a two-sided parametric t-test. In **a**, *p = 0.032 (+/+ vs. +/del52), 0.0183 (+/del52 vs. del52/del52), 0.0163 (del52/del52 vs. ins5/ins5); **p = 0.036 (del52/del52 vs. +/ins5); ***p = 0.0005 (+/+ vs. del52/del52) and in **b**, *p = 0.0126 (+/+ vs. ins5/ins5), 0.0238 (+/ins5 vs. ins5/ins5); ***p < 0.0001 except p = 0.0006 for del52/del52 vs. ins5/ins5. Source data are provided as a Source data file. **c** Histopathology of BM and spleen showing HES and RET stainings of ≥10 months post-tamoxifen mice. **d** HES staining showing MK hyperplasia and increase in red blood cells (RBCs) in spleen of homozygous del52/del52 mouse. Images were obtained using a DM2000 Leica microscope and a DFC300FX Leica camera with Leica Application Suite v.2.5,OR1 acquisition software (×10, ×25, and ×40 magnifications). Scale bars represent 50 µm. Similar results were obtained with analyzing at least three independent mice for each genotype.

**Role of homozygosity in the expansion of *CALR* ins5 HSCs**. On one hand, platelet counts were reported to be higher in ins5-ET than in del52-ET patients[16,17] and, on the other hand, homozygous clones are more frequently found in ins5- than in del52-mutated patients[2,3,18,19]. Thus, we decided to investigate how homozygous ins5 cells might out-compete heterozygous ins5 cells at the HSC level and how it would affect platelet levels knowing that thrombocytosis in homozygous ins5/ins5 mice was significantly superior than in heterozygous +/del52 mice (Fig. 8a). In order to test the effect of ins5 homozygosity on the severity of the disease, we performed a competitive transplantation of 20% of ins5/ins5 with 80% of +/ins5 BM cells into lethally irradiated recipient mice. One month after transplantation, KI expression was induced by tamoxifen. As expected, mice engrafted with 100% heterozygous del52 cells displayed higher platelet counts than 100% heterozygous ins5-transplanted mice, compared to engraftment control with 100% wt (+/+) cells (Fig. 8b). In the presence of limited homozygous ins5 cells with heterozygous ins5 cells, mice developed a significant stronger thrombocytosis than mice engrafted with 100% of heterozygous ins5 cells, and surpassed, but not significantly, the 100% heterozygous del52 model. Six months after KI induction, the percentage of ins5/ins5 HSCs (SLAM) after single SLAM cell sorting and genotyping of colonies had doubled compared to initially engrafted BM cell levels (Fig. 8c) showing that homozygous ins5 cells can outcompete heterozygous ins5 cells and may play an important role in the extent of thrombocytosis.

## Discussion

We report here a direct comparison of the effects of physiological expression of the two most frequent *CALR* mutations, del52 and ins5, on hematopoiesis. Del52 KI mice developed a more severe thrombocytosis than ins5 KI mice and in the homozygous than in the heterozygous status. The majority of platelet increase in homozygous del52 KI mice seems to be due to an increase in both the number and the size of MKs, while homozygous ins5 and heterozygous del52 thrombocytosis was mostly due to an increase in MK size, confirmed by an increased ploidy.

The extent of thrombocytosis developed by KI mice was related to the zygosity status of the mutations, as observed for del52 by Li et al.[13]. The strategies used to generate the two KI models were different but result in the same chimera sequence (Supplementary Fig. 10). This suggests that the magnitude of thrombocytosis is dependent on the amount of CALR mutants and probably on the ratio of CALR mutant to CALR wt. Of note, CALR del52 and CALR ins5 proteins were barely detectable in heterozygous KI cells, may be due to the role of CALR wt in the stability of the mutants and/or increased secretion. It is interesting to highlight that with 100% mutated HSCs, *CALR*-mutated KI mouse models are more closely related to patients than *JAK2*V617F KI mice because *CALR*-mutated, in contrast to *JAK2*V617F-patients, display a high allele burden (30–50%) at the HSC/progenitor level suggesting that most HSCs need to be mutated for disease development[20]. In contrast to our KI and RV mouse models[8],

levels of platelets in ins5-ET patients were reported to be higher than in del52-ETs[16,17]. Several hypotheses may explain the apparent difference in the levels of del52- versus ins5-induced thrombocytosis in patients compared to mice. First, in agreement with the observations performed in mice, one study reported that CD34+ progenitors transduced to express del52 generated more proplatelets than with ins5[21]. Second, in the mouse models, we can follow a time-dependent development of thrombocytosis, correlating with HSC/MkP amplification. Platelet counts start decreasing when del52 mice display signs of MF-like features. In both mice and patients, *CALR* mutations induce a continuum of diseases between ET, pre-fibrosis, and PMF with fluctuating platelet levels. While ins5 ETs present slightly higher platelet counts than del52 ETs, this difference is not further observed in MF[17]. In particular, del52 ETs are more inclined to progress towards MF than ins5 ETs, thus suggesting more heterogeneity in platelet counts for del52 patients. Although allele burden in *CALR*-mutated patients often tends to be close to 50% when disease declare, it would be interesting to compare platelet levels in del52 and ins5 ETs with similar and low allele burden and to follow progression of thrombocytosis to measure the real impact of these two mutations on platelet counts. Third, it was reported that homozygous *CALR* mutations, although they are rare, were rather ins5 and type 2-like than del52 and type 1-like mutations[2,3,18,19]. In ET patients, homozygous ins5 is frequently restricted to a small fraction of progenitors, but may contribute to high level of thrombocytosis. In favor of this hypothesis, when engrafting limited amount of homozygous ins5 cells with heterozygous ins5 cells in mice, the platelets counts undeniably exceeded levels reached by 100% heterozygous ins5-engrafted mice, and more interestingly, by 100% heterozygous del52-engrafted mice. In this model, the homozygous ins5 HSC reached 40% when thrombocytosis surpassed levels reached by the heterozygous del52 engrafted mice. In patients, the homozygous clone is lower, ranging from 2 to 25% in the CD34+ cell population, but it cannot be excluded that it is specifically amplified in the MK/platelet lineage. Thus, our study may provide a preliminary explanation on the apparent contrast between the del52 and ins5 thrombocytosis phenotype in mouse compared to human. Therefore, it will be important to correlate platelet counts with the percentages of homozygous *CALR*-mutated MKs or at least with the global *CALR*-mutated allele burden in MKs.

In contrast to *CALR*-mutated patients who show little leukocytosis compared to *JAK2*V617F NMPs, we observed that del52 and, to a slightly lesser extent, ins5 KI mice develop leukocytosis that is dependent on zygosity. In a similar manner, Li J et al also showed a slight increase in WBC in heterozygous del52 KI mice, more reproducible and significantly higher in homozygous animals[13]. Similarly to heterozygous del52 KI mice, there was a moderate leukocytosis in the CRISPR-Cas9-mediated murine del52 KI mice[12], while no change in WBC counts was seen in the human del52 TG and the CRISPR-Cas9-mediated murine del61 and murine del19 KI mice[10–12]. We show here that the increased WBC population in del52 and ins5 homozygous KI mice includes

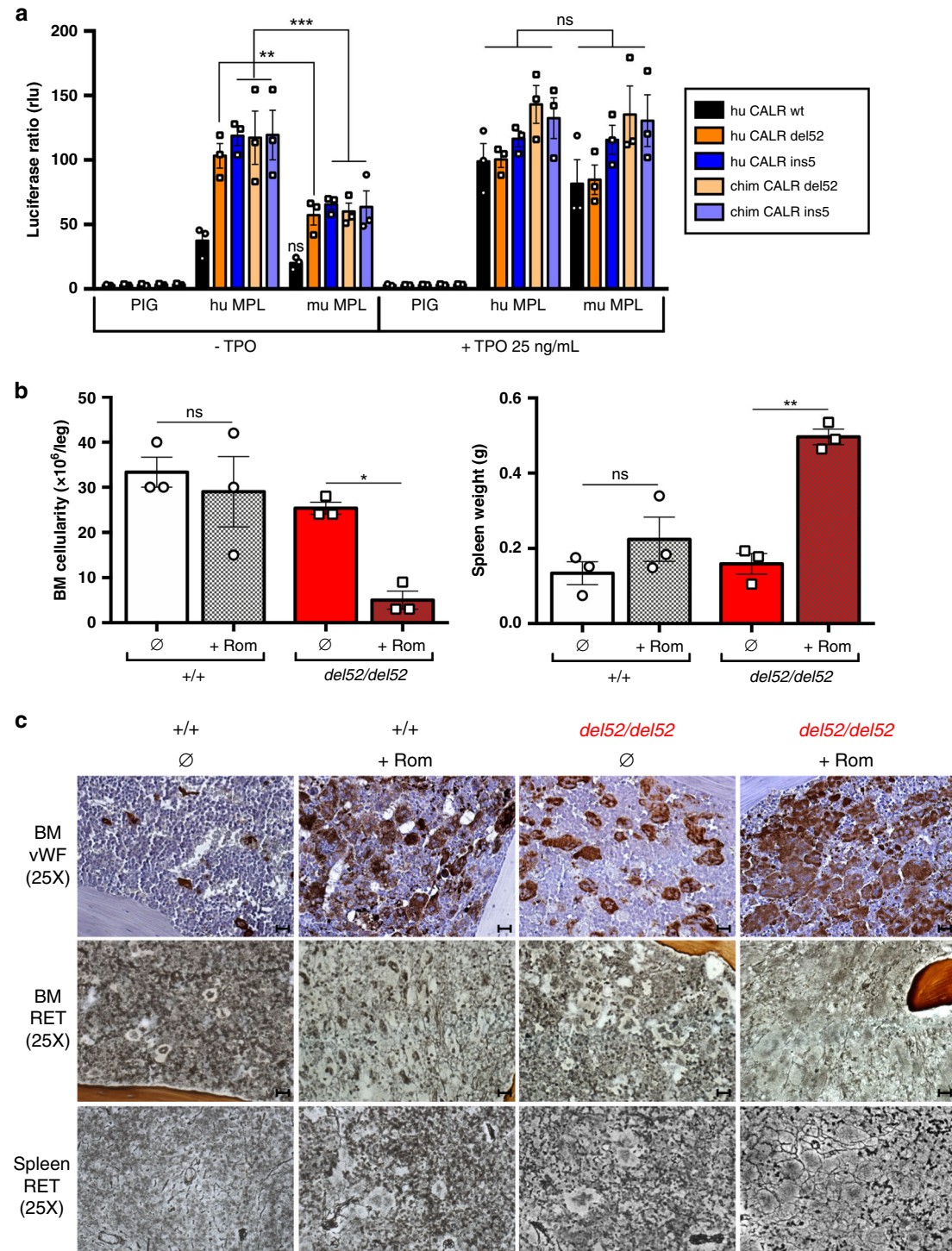

**Fig. 5 Degree of activation of MPL/JAK2/STAT determines the phenotype. a** The γ2A cells were transfected to express JAK2, mu MPL, hu MPL or an empty vector (PIG) and equal amounts of either the hu CALR wt (black), del52 (dark orange) or ins5 (dark blue) or the chim CALR del52 (light orange) or ins5 (light blue), as indicated. STAT5-dependent transcriptional activity with or without TPO was measured 24 h after transfection by the dual firefly luciferase system with Spi-Luc reporter (STAT5 response elements) and renilla pRL-TK reporter with constitutive expression, as an internal control. Shown are means ± SEM of three independent experiments in triplicate. Significance was assessed using Tukey's multiple comparison test (**p = 0.026; ***p = 0.0004 (hu CALR ins5), 0.0001 (chim CALR del52), 0.0002 (chim CALR ins5); ns not significant). **b** BM cellularity and spleen weight of 1 month post-tamoxifen *del52/del52* KI mice (red) and wild-type (+/+) littermates (white) treated with romiplostim (+Rom, hatched histograms) or not (∅, solid histograms). Data are means ± SEM (n = 3 different mice) and significance of +Rom to ∅ condition was assessed using a two-sided parametric paired *t*-test: *p = 0.0198; **p = 0.0017; ns: not significant. Source data are provided as a Source data file. **c** BM and spleen were examined after vWF immunostainning and RET staining. Images were obtained using a DM2000 Leica microscope and a DFC300FX Leica camera with Leica Application Suite v.2.5,OR1 acquisition software (×25 magnifications). Scale bars represent 50 μm. Similar results were obtained with analyzing three independent mice for each condition.

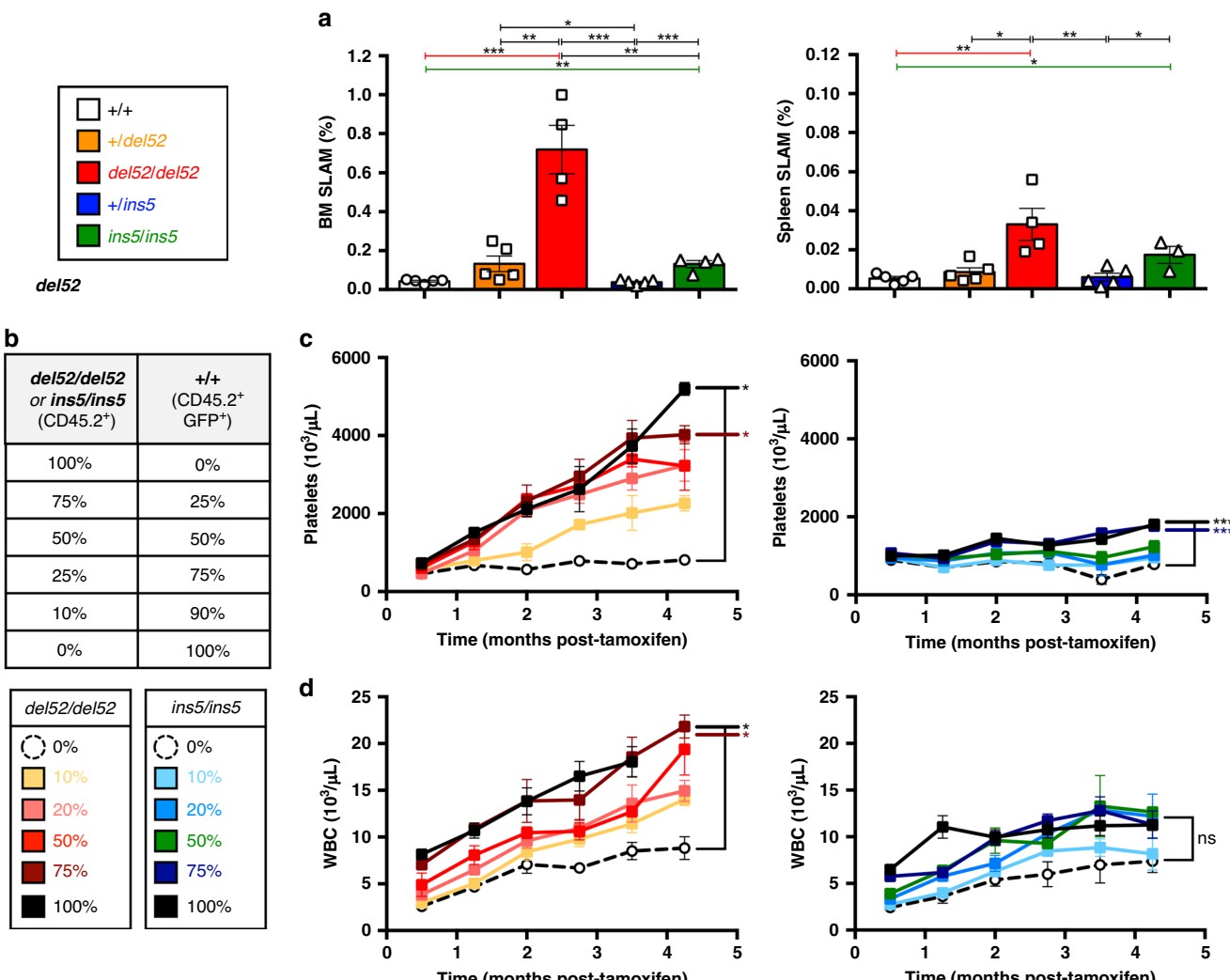

**Fig. 6 Effect of *CALR del52* and *CALR ins5* on the HSCs. a** Percentages of HSCs (SLAM, Lin⁻c-Kit⁺Sca-⁺CD150⁻CD48⁺) in BM and spleen of ≥10 months post-tamoxifen mice were expressed as means ± SEM ($n = 5$ independent mice of +/+ (white), +/del52 (orange) and +/*ins5* (blue) genotypes; $n = 4$ *del52/del52* (red) mice and for *ins5/ins5* (green) genotype $n = 4$ for BM and $n = 3$ for spleen analyses). Significance was determined using a two-sided parametric *t*-test: in BM, *$p = 0.047$; **$p = 0.0012$ (+/+ vs. ins5/ins5), 0.0016 (+/del52 vs. del52/del52), 0.0034 (del52/del52 vs. ins5/ins5); ***$p = 0.0004$ except 0.0009 for +/*ins5* vs. ins5/ins5 and in spleen, *$p = 0.0157$ (+/del52 vs. del52/del52), 0.0131 (+/+ vs. ins5/ins5); **$p = 0.0071$ (+/+ vs. del52/del52), 0.0094 (del52/del52 vs. +/ins5), 0.0334 (+/ins5 vs. ins5/ins5). **b** To investigate the effect of *CALR del52* and *CALR ins5* on HSCs we engrafted different ratios of homozygous *del52/del52* (yellow, pink, red, brown and black squares and solid lines) and *ins5/ins5* (turquoise, blue, green, dark blue and black squares with solid lines) CD45.2⁺ KI BM cells with wild-type (+/+) CD45.2⁺ GFP⁺ BM cells in CD45.1⁺ lethally irradiated mice. Four weeks after transplantation, KI allele expression was induced by tamoxifen and **c** platelet and **d** WBC counts were measured. Data are the means ± SEM, *del52/del52* platelets: $n = 4, 4, 2, 4, 4, 3$ for 0%, $n = 4, 4, 3, 3, 3, 3$ for 10%, $n = 5, 4, 5, 4, 4, 4$ for 25%, $n = 5, 5, 4, 5, 5, 4$ for 50%, $n = 3, 4, 4, 3, 4, 3$ for 75% and $n = 5, 5, 5, 3, 5, 2$ for 100%; *ins5/ins5* platelets: $n = 3, 3, 3, 1, 3, 3$ for 0%, $n = 5, 4, 3, 4, 2, 4$ for 10%, $n = 5, 5, 4, 4, 2, 3$ for 25%, $n = 5, 5, 3, 4, 3, 5$ for 50%, $n = 5, 5, 3, 4, 3, 5$ for 75% and $n = 5, 5, 3, 4, 4, 5$ for 100%; *del52/del52* WBC: $n = 4, 4, 3, 4, 4, 4$ for 0%, $n = 4$ for 10%, $n = 5, 4, 5, 5, 4, 5$ for 25%, $n = 5, 5, 5, 5, 5, 4$ for 50%, $n = 4, 4, 4, 3, 4, 4$ for 75% and $n = 5, 5, 5, 4, 5$ for 100%; *ins5/ins5* WBC: $n = 3, 3, 3, 2, 3, 3$ for 0%, $n = 5, 5, 4, 4, 5, 5$ for 10%, $n = 5, 5, 5, 5, 3, 4$ for 25%, $n = 5, 5, 5, 5, 3, 5$ for 50%, $n = 5, 5, 3, 5, 4, 5$ for 75% and $n = 5, 4, 5, 5, 5, 5$ for 100%. *N* represents individual mice and varies depending on time of sampling during the course of the experiment and on the group defined as the percentage of engrafted homozygous cells, as indicated in the Source data file. Significance compared to the 0% condition (open circles with connecting black dashed line) was determined using ANOVA with Dunnett's multiple comparison test and a nonparametric Kruskal–Wallis's multiple comparison test for *ins5/ins5* WBC. In **c**, *$p = 0.0172$ (100%), 0.0256 (75%); ***$p = 0.0001$ (100% and 75%) and in **d**, *$p = 0.0422$ (100%), 0.0128 (75%); ns not significant. Source data are provided as a Source data file.

neutrophils, B and T lymphocytes, natural killer (NK) cells and eosinophils as well as monocytes and basophils that were differentially regulated by CALR del52 and ins5. The increase in neutrophils in *del52* KI mice could be the consequence of activation of the G-CSF receptor by CALR mutants[5]. Moreover, as homozygous *ins5* patients were reported to present a deficiency in both the neutrophil myeloperoxidase (MPO) and the eosinophilic

peroxidase (EPX)[22], it could be expected to see neutropenia in mice placed in a pathogen-challenged environment. Of particular interest, our KI models present high levels of circulating lymphocytes, especially of various sub-types of T cells. However, it is difficult to know whether this increase results from CALR mutant immunogenicity and the activation of lymphocytes. This hypothesis is a matter of debate as work from collaborators shows

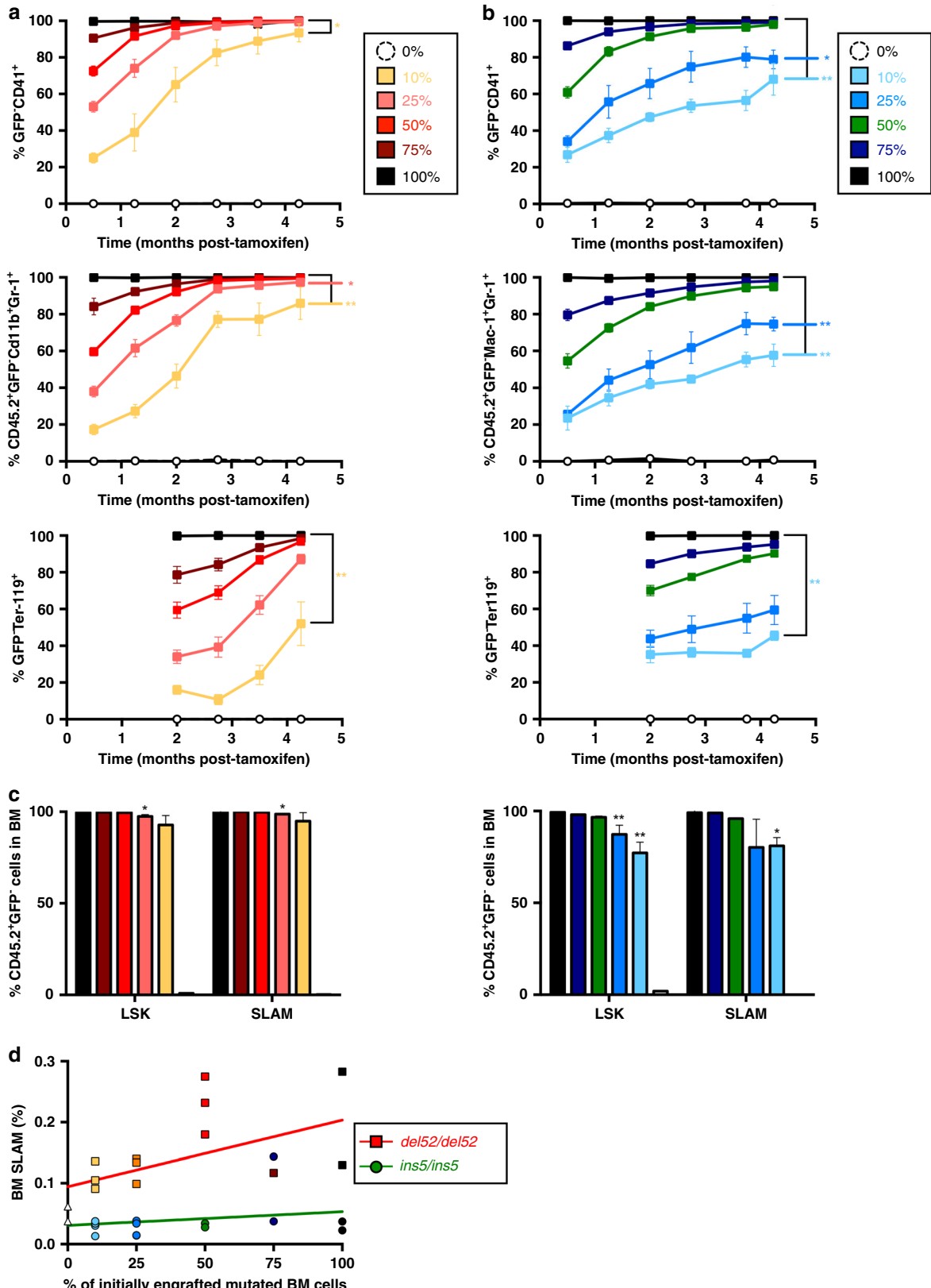

that CALR del52 has an immunosuppressive effect[23] and causes a down-regulation of MHC-I endogenous antigen loading and presentation[24] but CALR mutant epitopes were also shown to stimulate CD4[+] and CD8[+] T cells in patients[25,26]. Importantly, in contrast to *JAK2*[V617F], which is detected at a low level in lymphoid cells and rarely in T lymphocytes from MPN patients,

*CALR* mutations are present in the majority of monocytes and NK cells and frequently detected in T lymphocytes, most likely owing to the effect of both *del52* and *ins5* on HSC expansion. Thus, the presence of *CALR* mutations in lymphoid cells might also have a direct effect on their expansion or their blood trafficking that should be investigated in the future.

**Fig. 7 Evolution of homozygous del52 and ins5 cells in blood and BM cells.** Percentages of **a** del52/del52 (yellow, pink, red, brown and black squares and solid lines) and **b** ins5/ins5 (turquoise, blue, green, dark blue and black squares with solid lines) mutated KI cells in platelets (CD41$^+$) and erythroid (Ter-119$^+$) blood cells was determined as GFP$^-$ cells and in granulocytes (Cd11b$^+$Gr-1$^+$) as CD45.2$^+$GFP$^-$ cells, and are expressed as means ± SEM (in **a**: $n = 4$ for 0%, $n = 4$ GFP$^-$CD41$^+$, $n = 4$, 4, 4, 2, 4, 4 CD45.2$^+$GFP$^-$Cd11b$^+$Gr-1$^+$, $n = 4$, 4, 3 GFP$^-$Ter-119$^+$ for 10%, $n = 5$ for 25 and 50%, $n = 4$, 4, 4, 4, 3, 4 GFP$^-$CD41$^+$ and $n = 4$ for 75%, $n = 5$ and 2 for the last point for 100%; in **b**: $n = 3$ for 0%, $n = 4$ for 10%, $n = 4$ and 3 for the last point for 25%, $n = 5$ for 50%, 75%, and 100%, with $n$ being individual mice and varying depending on time of sampling during the course of the experiment and on the group defined as the percentage of engrafted homozygous cells, as indicated in the Source data file). Significance to the 100% condition (black squares with connecting black solid line) was measured with nonparametric Kruskal–Wallis's multiple comparison test. In **a**, *$p = 0.0167$ (10%, GFP$^-$CD41$^+$), 0.0334 (25%, CD45.2$^+$GFP$^-$Cd11b$^+$Gr-1$^+$); **$p = 0.0023$ (10%, CD45.2$^+$GFP$^-$Cd11b$^+$Gr-1$^+$), 0.0094 (10%, GFP$^-$Ter-119$^+$) and in **b**, *$p = 0.0106$ (25%, GFP$^-$CD41$^+$); **$p = 0.0011$ (10%, GFP$^-$CD41$^+$), 0.0098 (25%, CD45.2$^+$GFP$^-$Cd11b$^+$Gr-1$^+$), 0.0015 (10%, CD45.2$^+$GFP$^-$Cd11b$^+$Gr-1$^+$), 0.0081 (10%, GFP$^-$Ter-119$^+$). **c** Percentages of homozygous del52 and ins5 cells (CD45.2$^+$GFP$^-$ cells) were assessed in HSC-enriched BM LSK and SLAM cells and are shown as means ± SEM, significance to the 100% condition with a two-sided parametric $t$-test: *$p = 0.0344$ (25% del52/del52, LSK), 0.0152 (del52/del52 SLAM), 0.0132 (10%, ins5/ins5 SLAM); **$p = 0.0143$ (25%, ins5/ins5 LSK), 0.0028 (10%, ins5/ins5 LSK). **d** The percentages of BM SLAM 4 months after tamoxifen induction of KI expression were determined and expressed according to the percentages of initially engrafted homozygous del52 (colored squares and red solid line) and homozygous ins5 (colored circles and green solid line) BM cells. For **c**, **d**, $n = 1$–5 independent mouse BM were analyzed, depending on the percentage of engrafted homozygous cells, as indicated in the Source data file. Source data are provided as a Source data file.

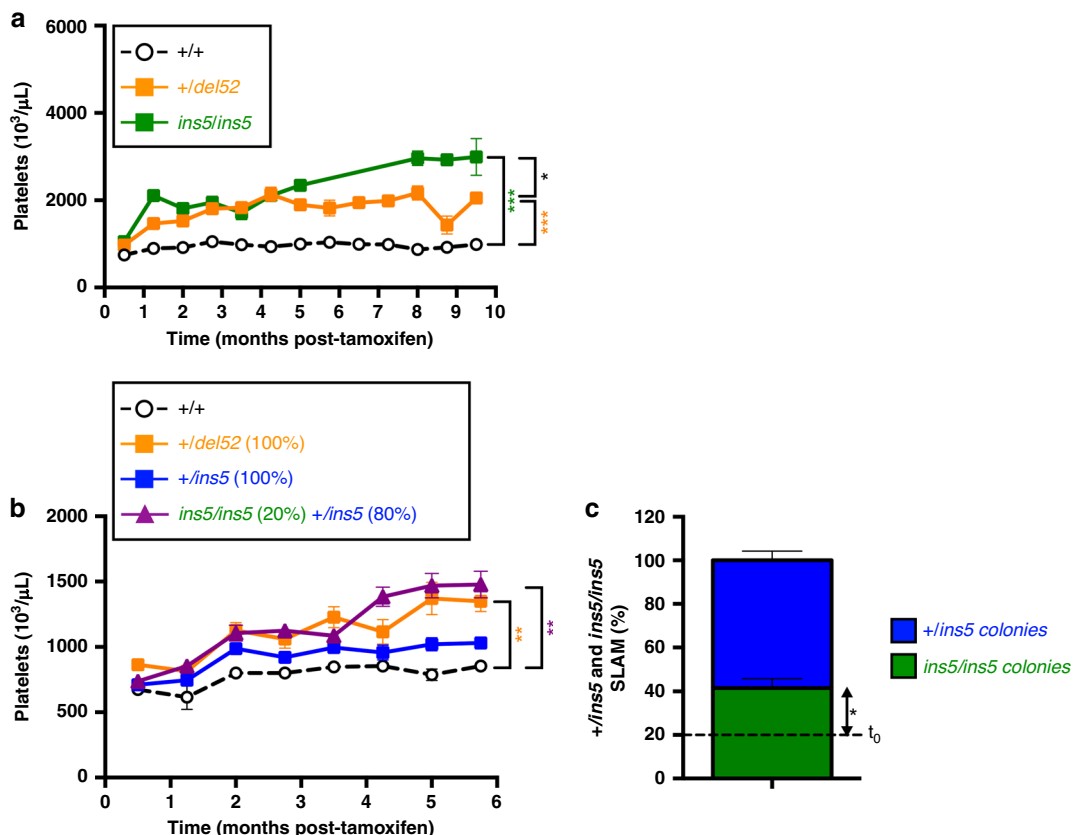

**Fig. 8 Homozygous ins5 cells drive the severity of the disease phenotype. a** Comparison of the time-dependent increase in blood platelet levels of heterozygous +/del52 (orange squares and solid line) and homozygous ins5/ins5 (green squares and solid line) KI mice after tamoxifen induction. Means ± SEM ($n = 16$, 18, 15, 11, 11, 8, 13, 7, 14, 11, 5, 6, 4 +/+, $n = 18$, 20, 15, 12, 10, 7, 11, 7, 12, 12, 8, 5, 10 +/del52 and $n = 6$, 1, 6, 8, 9, 7, 5, 6, 5, 5 ins5/ins5 independent mice, as shown in the Source data file) and significance was determined using ANOVA with Tukey's multiple comparison test (*$p = 0.0365$; ***$p < 0.001$). To investigate the effect of homozygous ins5/ins5 cells in the development of thrombocytosis we engrafted 20% of ins5/ins5 KI BM cells with 80% of heterozygous +/ins5 KI BM cells in lethally irradiated mice (purple triangles with solid line). As controls we also engrafted 100% wt (+/+, open circles with black solid line), heterozygous +/del52 orange squares with solid line) or +/ins5 (blue squares with solid line) KI BM cells. Four weeks after transplantation, KI allele expression was induced by tamoxifen and **b** platelets were measured. Figure shows the means ± SEM ($n = 6$ +/+ $n = 8$, 8, 8, 8, 7, 8, 8, 8 +/del52, $n = 8$ and 7 for the last point +/ins5 and $n = 8$, 8, 8, 8, 3, 8, 7, 7 ins5/ins5 and +/ins5 mixed individual mice, as indicated in the Source data file). Significance was calculated with a nonparametric Kruskal–Wallis's multiple comparison test (**$p = 0.0076$ (+/+ vs. 20% ins5/ins5), 0.0083 (+/+ vs. 100%.+/del52). **c** At 6 months post-induction, single BM SLAM were sorted for a 10-day culture with all cytokines. Colonies were genotyped to retrospectively identify percentages of ins5/ins5 (green) and +/ins5 (blue) SLAMs. Data are the means ± SEM ($n = 3$ mice with an average of 58 genotyped colonies/mouse). Significance compared to the 20% of initially engrafted ins5/ins5 BM cells (dashed line, $t_0$) was measured with a two-sided parametric $t$-test (*$p = 0.0492$). Source data are provided as a Source data file.

Homozygous *del52* and to a lesser extent homozygous *ins5* and heterozygous *del52* led to the amplification of the HSC compartment in BM and spleen. Competitive BM engraftment of homozygous *del52* and *ins5* cells confirmed that *del52* activity is more potent than *ins5* in promoting an advantage over the wt hematopoiesis, as previously shown in the RV model[8]. Indeed, an increase in the percentage of *ins5*-mutated cells was very slow when initial engraftment was below 50% of mutated BM cells, especially in the RBC and granulocyte compartments but also in platelets and in the BM HSC compartment. As mentioned earlier, it was observed that homozygous clones are more often identified in *ins5*-patients than in *del52*-patients, suggesting the importance of a gene-dosage for *ins5* oncogenic activity. In contrast, an absence of growth advantage of *del52* compared to *CALR wt* was observed using the *del52* TG BM cells in an initial 50% competitive transplantation[10]. Similarly, the effect of *del52* on HSC was absent when Li et al. performed a 50% homozygous *del52* BM cell competitive transplantation[13]. In these two studies, the heterozygous/homozygous *del52* expression was induced in donor mice before engraftment while we engrafted wt cells and induced homozygous del52 KI expression 1 month later. The only difference between the models of the group of AR Green and of our laboratory is the type of transgenic mice that were crossed with the floxed KI mice, Mx1-Cre (ubiquitous expression), and SCL-CreER$^T$ (HSC and endothelial cell expression), respectively. Thus, our hypothesis is that either del52-expressing cells display a homing defect and/or that expression of del52 in the Mx1-Cre context is not sufficient to induce a competitive advantage. Alternatively, it is also possible that the amplification of HSCs observed in KI mice may correspond to an increase of a subpopulation of active HSCs with more proliferating capacities, that would result in milder secondary engraftment. Interestingly, the group of K Shimoda showed that CALR wt is able to inhibit the CALR del52-induced competitive advantage, thus it is possible that the total absence of CALR wt in homozygous *del52* or *ins5* HSCs might explain their advantage on their heterozygous counterparts[27]. With age, preferentially homozygous *del52* KI mice developed splenomegaly accompanied with a decrease in BM cellularity, while blood parameters remained constant. A decrease in BM cellularity was partially explained with the striking MK hyperplasia and increase in MK ploidy, as well as the decrease in the erythroid lineage while the granulocytic lineage remained constant. Splenomegaly was due to extramedullary hematopoiesis characterized with an increased frequency of giant MKs, erythroid, and granulocytic cells at the expense of the lymphoid cell populations. There was no global decrease in RBC since the decrease in BM was compensated by splenic erythropoiesis. Surprisingly, while the role of MKs in the development of MF has been largely documented[28], heterozygous *del52* KI mice, like human *del52* TG and murine *del19*, *del52*, and *del61* KI mice, did not develop or only minor fibrosis even in aged animals, although they present increased numbers of MKs in both BM and spleen. The homozygous *del52* KI mice, that presented an even higher MK hyperplasia in BM and spleen, showed minor fibrosis in BM and moderate fibrosis in the spleen. However, this difference in the grades of fibrosis severity seemed to be independent of different signaling (STAT5) intensities in BM *versus* spleen MKs. Conversely, moderate fibrosis in BM but not in the spleen was observed in the homozygous *del52* KI mice of Li et al. while MK hyperplasia was observed in both tissues[13]. Moreover, the progression of homozygous *ins5* KI mice to fibrosis was observed in rare cases with a lower grade compared to homozygous *del52* KI mice, maybe due to the highest amplification of MKs in the *del52* setting. These observations are coherent with the RV models where del52-compared to ins5-expressing mice develop high-grade fibrosis in both BM and spleen from 6 months post-BM transplantation[8]. We hypothesized that the differences of fibrosis in the RV compared to the KI models could be due to the activation of mu MPL by hu

CALR mutants that were reported to be more efficient than the mouse/mouse combination[11,12]. As our KI mice express a murine CALR with a human mutated C-terminal tail chimeric protein, we tested to which extent they could activate mu MPL, compared to hu CALR del52 and ins5. The chimeric proteins were as efficient as hu CALR mutants to activate mu MPL, but chimeric protein and hu CALR were less active towards mu MPL than hu MPL. When MPL signaling in the homozygous *del52* KI mice was stimulated by the TPO agonist romiplostim, fibrosis was visible in both BM and spleen at slightly higher levels than wt-treated littermates. Thus, the suboptimal activation of MPL by the chim CALR del52 and chim CALR ins5 in our KI mice, that was confirmed by the over-activation of STAT5 by in vitro TPO stimulation of MK might partially explain the mild fibrosis phenotype. Of note, we did not detect any significant difference between CALR del52 and CALR ins5-induced activation of STAT5 in γ2A cell line. Whether this activation is also similar in vivo in HSCs or not and whether other signaling pathways may contribute to the distinct phenotypes developed by del52 and ins5 KI mice remain to be tested.

Overall, our KI mouse models show a differential effect of the two most frequent *del52* and *ins5* CALR mutations in mouse hematopoiesis, in both a heterozygous and a homozygous setting. Therefore, these KI mice provide a powerful tool to better understand the differences between *del52* and *ins5* CALR mutations and to perform preclinical studies. In addition, our observations suggest that homozygous *ins5* clone may play a more important role in the ET pathogenesis than previously thought, giving a future avenue of research in MPN pathogenesis.

## Methods

**Generation of conditional heterozygous and homozygous *CALR del52* and *CALR ins5* KI mice**. *CALR del52* and *ins5* KI mice were generated at the Institut Clinique de la Souris (ICS, Illkirch, France). The targeting vectors were derived from a genomic DNA fragment of murine *Calr* gene obtained from 129S2/SvPas mouse genomic DNA library. The intron 8 of the gene was modified to integrate firstly a cassette containing a fused exon 8⁻exon 9 with the stop site and a triple polyA sequence surrounded with two LoxP sites, followed by a *Pgk-NeoR* cassette flanked by FRT sites and, secondly, a modified exon 9 with the 52-bp deletion (*del52*) or 5-bp insertion (*ins5*) mutation (untranslated). Due to the lack of homology between human and mouse in the alternative reading frame of the 3′UTR induced by the frameshift of the mutations, we chose an alternative strategy to express the putative *del52* and *ins5 Calr*-mutated murine genes by expressing a chimeric murine/human protein with most of the murine gene (except from the 52-bp deletion or 5-bp insertion in the last exon 9) and part of the 3′UTR human *CALR*, translated by the frameshift mutations. The human sequence was: AGGAT GATGAGGACAAAGATGAGGATGAGGAGGATGAGGAGGACAAGGAGGAA GATGAGGAGGAAGATGTCCCCGGCCAGGCCAAGGACGAGCTG**TAG**AGA GGCCTGCCTCCAGGGCTGGACTGAGGCC**TGA**, the human translated part after the mutated mouse CALR protein being: RRMMRTKMRMRRMRRTRRK MRRKMSPARPRTSCREACLQGWTEA.

The linearized homologous recombination vector constructs, including ≈4 kb 5′ and 3′ homology arms, were electroporated into C57BL/6N embryonic stem (ES) cells followed by G418 selection. Polymerase chain reaction (PCR) and Southern blot were used to screen and identify homologously recombined ES cells that were injected into blastocysts and reimplanted into the uterine horns of pseudo-pregnant foster females to allow embryos to develop to term. Chimeras were crossed with C57BL/6N Tac mice carrying the *Flp* recombinase transgene in order to excise the *Pgk-NeoR* cassette[29]. Agouti color hair progenies were genotyped for germinal transmission of the targeted allele using PCR and Southern blotting analyses. Subsequently, these LoxP KI mice were transferred, bred, and studied in the animal facilities of Gustave Roussy. Cre recombination after crossing the LoxP KI mice with Cre-expressing transgenic mice (HSC-SCL-Cre-ER$^T$)[30] results in excision of the fused normal exons and expression of the genes with the mutated exon 9. Three-month-old Cre/LoxP KI mouse progenies were treated with tamoxifen (200 mg/kg daily dose, Sigma-Aldrich) for 4 consecutive days.

**Mouse genotyping**. Conditional *del52* KI mice were genotyped after genomic DNA extraction using the Mouse Direct PCR kit (Biotool) with a and b primers generating the wt (503 bp) and a conditional KI (404 bp) allele bands. The conditional *ins5* KI mice genotyping was similarly performed by PCR with the primers c and d amplifying the wt (217 bp) and the conditional KI (297 bp) alleles. The presence of the SCL-CreER$^T$ copies was detected by PCR with p1, p2, p3, and p4 generating a 700 bp wt band (p1 and p2) and a 1,100 bp SCL-CreER$^T$ band (p3 and

p4). Primer sequences are listed in Supplementary Table 1. Gel imaging was analyzed with Image Lab software (Bio Rad).

**Analysis of mice.** Animal experiments were conducted in Gustave Roussy animal facility and approved by Gustave Roussy review board, protocol no. 2016-104-7171. Mice were housed under 12-hour light/12-hour dark cycles and monitored ambient temperature (21 ± 1 °C) and humidity range (20–70%).

**Blood, BM, and spleen.** Platelet, hemoglobin, and WBC counts were determined using an automated counter (MS9; Schloessing Melet). BM cells were collected by flushing the femurs and tibias. Spleens were weighed and single-cell suspensions were prepared.

**Content in progenitor and precursor cells by flow cytometry.** Flow cytometry (LSRFortessa, FACSCanto X, BD Biosciences) with BD FACSDiva software was used to determine the cell content of BM and spleen with appropriate Abs. Lin$^-$ cells were selected on the absence of labeling with APC-conjugated rat Abs against Ter-119, B220, Gr-1, CD3, and CD11b. Lin$^-$Sca-1$^+$c-Kit$^+$ (LSK) population was stained with Abs against Sca-1 (PE-Cy7) and c-Kit (PerCP-Cy5.5) and the LSKCD48$^-$CD150$^+$ (SLAM) population enriched in HSCs was isolated with anti-CD48 (Pacific Blue) and CD150 (BV510) Abs. Anti-CD41 (FITC) Ab was used to label MK progenitor (MkP) cells with the LKSca-1$^-$CD150$^+$CD41$^+$ combination. BM and spleen precursors were labeled as CD41$^+$CD42$^+$ (MK), Cd11b$^+$Gr-1$^+$ (granulocyte lineage), CD3$^+$ and B220$^+$ (T and B cells) and Ter-119$^+$ (erythroid cells). All Abs were obtained from BioLegend (Ozyme). Data were analyzed with Kaluza Analysis software (Beckman Coulter).

**Analysis of WBC populations by flow cytometry.** To investigate the cell population composition of WBCs (gating strategy in Supplementary Fig. 2a), we labeled blood using anti-Ly6G (APC-Cy7) Ab to identify polynuclear neutrophils (Ly6G$^+$). In the Ly6G$^-$ cell population we labeled B and T lymphocytes as B220$^+$ (BV510) and CD3$^+$ (PB), respectively. Among the Ly6G$^-$B220$^-$CD3$^-$ cells, NK cells were stained as NKp46$^+$ (FITC) while monocytes were stained as CD11b$^+$ (PE) Ly6C$^{-/+}$ (APC) in the Ly6G$^-$B220$^-$CD3$^-$NKp46$^-$ cell population. Gating on Ly6G$^-$B220$^-$CD3$^-$NKp46$^-$ CD11b$^-$cells we used anti-CD45 (PE-Texas Red) Ab and evaluated granularity with the sidewards scatter (SSC) parameter to distinguish eosinophils (CD45$^{high}$SSC$^{high}$) from basophils (CD45$^{low}$SSC$^{low}$). We further investigated different lymphocyte sub-populations (gating strategy in Supplementary Fig. 3a) in the Ly6G$^-$ cell population by combining Abs to detect T-cytotoxic lymphocytes as CD3$^+$ (Pacific Blue) CD4$^-$ (APC) CD8$^+$ (PE) and T-helper lymphocytes (TH) as CD3$^+$CD4$^+$CD8$^-$. In the TH gate, we used anti-FoxP3 (Alexa Fluor 488) and anti-CD25 (BV510) Abs to stain regulatory CD4$^+$ T cells (Tregs, CD3$^+$CD4$^+$CD8$^-$FoxP3$^+$CD25$^{+/-}$) and conventional CD4$^+$ T cells (TC, CD3$^+$CD4$^+$CD8$^-$FoxP3$^-$CD25$^-$). All Abs and the FoxP3 Fix/Perm buffer set were from BioLegend. Data were acquired with BD FACSDiva software and analyzed with Kaluza Analysis software (Beckman Coulter).

**Progenitor assays in semi-solid cultures.** Spleen and BM cells were plated in methylcellulose MethoCult 32/34 (Stemcell Technologies) with 10 ng/mL hu TPO (a generous gift from Kirin, Tokyo, Japan), 1 U/mL hu EPO (PreproTech), 10 ng/mL mu IL-3 (Miltenyi Biotec), 10 ng/mL hu IL-6 (Miltenyi Biotec) and 100 ng/mL mu SCF (PreproTech). After 7 days, colonies derived from erythroid (BFU-E, burst forming unit-erythroid) and granulo-monocytic (CFU-GM, colony forming unit-granulocyte macrophage) progenitors were counted. For megakaryocytic progenitor (CFU-MK) assay, BM nucleated cells were plated in duplicate in serum-free fibrin clot assays with 25 ng/mL mu SCF, 25 ng/mL hu IL-6 and without or with 20 ng/mL hu TPO. CFU-MKs were evaluated at day 7 by acetylcholinesterase staining.

**Histology of BM and spleen.** Tibia and spleen sections were fixed in 4% paraformaldehyde, decalcified, and paraffin-embedded. Sections were stained with HES for cytology and MKs were immune-stained with rabbit anti-vWF Ab. Reticulin fibers were revealed by Gordon and Sweet's silver staining.

**Romiplostim treatment.** One month after tamoxifen induction of KI allele expression, homozygous del52/del52 KI mice and wt (+/+) littermates were administered 1 mg/kg romiplostim (Nplate, Amgen) by subcutaneous injection on days 1, 8, and 15 based on a published study[15]. Platelets, WBC and hemoglobin were recorded on days 1 and 15. At day 20 mice were sacrificed to determine BM cellularity, spleen weight and the presence of reticulin fibers (Gordon and Sweet's silver staining) and MKs (vWF immunohistochemical staining) in BM and spleen.

**Ploidy of BM MKs.** BM cells were labeled with FITC-conjugated anti-CD41 and APC-conjugated anti-CD42 Abs (BioLegend). After hypotonic cell permeabilization with 0.1% sodium citrate, DNA content was stained with 50 μg/mL propidium iodide (Sigma) and treated with 50 μg/mL RNAase (Sigma). Ploidy was measured by flow cytometry in the CD41$^+$CD42$^+$ cell population by means of a LSRFortessa

(Becton Dickinson) with a BD FACSDiva software. Data were analyzed with Kaluza Analysis software (Beckman Coulter). The mean ploidy was calculated in ≥8N MKs ((8N × number of cells at 8N ploidy level +…+64N × number of cells at 64N ploidy level)/total number of cells).

**Western blot analysis.** Lin$^-$ cells isolated from BM were analyzed by western blot to determine the expression levels of CALR using a rabbit monoclonal Ab directed against the N-terminal portion of CALR (Cell Signaling, Ozyme), common to wt as well as mutant CALR. An Ab directed against amino acids 50–150 of human CALR from Abcam was also used, as well as a mouse monoclonal Ab specifically directed against the C-neoterminus of mutant CALR (Dianova). Mouse monoclonal anti-β-Actin (Sigma) served to monitor the loading. Blot was quantified using ImageJ software (National Institute of Health).

**Dual luciferase transcriptional assay.** The KI allele encodes the mouse CALR ins5 or del52 with human mutated C-terminal tail. These chimeric CALR del52 and CALR ins5 cDNAs were amplified by PCR from homozygous del52 and ins5 KI BM cells and introduced into a pMSCV-IRES-GFP vector. Transcriptional activation of STAT5 was analyzed in γ2A cells using the dual firefly reporter Spi-Luc and pRL-TK-driven renilla luciferase system (Promega). γ2A cells were transiently transfected using Lipofectamine 2000 reagent (ThermoFisher) with cDNAs coding for mu STAT5, mu JAK2 as the cells are defective in JAK2, hu CALR wt, hu CALR del52, hu CALR ins5, chim CALR del52 or chim CALR ins5 in the presence of mu MPL, hu MPL, or the empty pMX-IRES-GFP (PIG) vector, as indicated. Cells were stimulated or not with 25 ng/mL TPO (Miltenyi Biotec) for 24 h.

**Phosphoflow assay.** Lin$^-$ cells isolated on ice from BM and spleen from ≥10 months post-tamoxifen del52/del52 KI mice and from wt (+/+) littermates were stimulated or not with 100 ng/mL TPO for 15 min or incubated with 1 μM ruxolitinib (Selleck Chemicals, Euromedex) for 1 h at 37 °C, as indicated. Cells were fixed with 1.6% formaldehyde (ThermoFisher) and MKs were stained with Abs against CD41 (Alexa Fluor 700) and CD42d (APC) from BioLegend. After methanol permeabilization and staining for p-STAT5 (BV421, BD Biosciences), samples were analyzed by flow cytometry (LSRFortessa) with BD FACSDiva software. Data were analyzed with Kaluza Analysis software (Beckman Coulter).

**Competitive transplantation models.** All procedures were approved by the Gustave Roussy Ethics Committee (protocol 2016-104-7171). Lethally irradiated CD45.1$^+$ wt recipient mice (C57BL/6LY5.1, Charles River) were engrafted with 3.10$^6$ of different ratios (100%; 75%; %; 25%; 10 and 0%) of BM cells from CD45.2$^+$ floxed homozygous del52 or ins5 KI mice with transgenic CD45.2$^+$ wt mice expressing GFP (UBI-GFP/BL6)[31]. Four weeks after transplantation, KI allele expression was induced by tamoxifen. Blood parameters were followed every 3 weeks and the competitive advantage of the mutated del52/del52 or ins5/ins5 cells in the blood was monitored by flow cytometry as GFP$^-$CD41$^+$ in platelets, GFP$^-$Ter-119$^+$ in erythrocytes and as CD45.2$^+$GFP$^-$Cd11b$^+$Gr-1$^+$ in granulocytes. At 4 months after induction, the percentages of mutated cells were analyzed in LSK/SLAM compartments of both BM and spleen.

A competitive ins5 KI transplantation model was made by engrafting 3.10$^6$ of a limited number of floxed homozygous ins5/ins5 KI BM cells (20%) in presence of floxed heterozygous +/ins5 KI cells (80%) into lethally irradiated wt recipient mice (C57BL/6JRj, Janvier Labs). As controls, 100% of floxed +/ins5 KI, +/del52 KI, and +/+ BM cells were also engrafted, as indicated. After 4 weeks, KI allele expression was induced by tamoxifen and blood parameters were analyzed every 3 weeks. At 6 months post-induction, SLAM cells were labeled from Lin$^-$ cells isolated from BM and sorted (Influx, Becton Dickinson) at 1 cell per well in a 96-well plate. After 10 days of culture in a serum-free medium supplemented with 10 ng/mL hu TPO, 1 U/mL hu EPO, 20 ng/mL hu G-GSF (a generous gift from Gustave Roussy Hospital), 10 ng/mL mu IL-3, 10 ng/mL hu IL-6, 100 ng/mL mu SCF, and 100 ng/mL hu FLT-3L (Miltenyi Biotec), colonies were lysed to recover genomic DNA for PCR genotyping to retrospectively identify heterozygous and homozygous ins5 KI SLAM (Supplementary Fig. 11). Primers used to detect the wt mouse Calr allele (302 bp) were muCALR_wt_ex8_F and muCALR_wt_onT2_R. Primers used to detect the ins5 KI allele (177 bp) were muCALR_wt_int8-9_F and chimCALR_ins5_R. All primer sequences are listed in Supplementary Table 1.

**Statistical analysis.** Statistical analyses were performed using the Prism software. If not otherwise mentioned, tests used an unpaired design. Comparisons were made using a parametric two-tailed t-test or a nonparametric Mann–Whitney test when needed, and ANOVA with Tukey's or Dunnett's multiple comparison tests or nonparametric Kruskal–Wallis multiple comparison test, as indicated. Differences were significative for $p < 0.05$.

## Data availability
The authors declare that all relevant data supporting the findings of this study are included in the paper and its supplementary information files. Source data are provided with this paper.

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

## Acknowledgements

This work was supported by grants from Ligue Nationale Contre le Cancer ("Equipe labellisée 2018", HR); Institut National du Cancer (PLBIO-2015, -2017 and -2018, IP), MPN Research Foundation (IP and JLV) and Institut National de la Santé et de la Recherche Médicale (Inserm). The "Investissements d'avenir" program is funding the Labex GR-Ex (IP, WV). MCC was funded by São Paulo Research Foundation 2016/03265-3. We thank O. Bawa for histology, the staff of the animal facilities of Gustave Roussy and C. Catelain and Y. Lecluse at the Imaging and Cytometry Platform. We thank M-C. Birling and E. Geronimus at the Institut Clinique de la Souris (Illkirch) for the generation of KI mice.

## Author contributions

C.M., I.P., and W.V. conceived and designed the study, interpreted the data and wrote the paper. C.M., C.B., and M.C.C. designed and performed experiments, analyzed the data and prepared the figures. A.N. performed and analyzed luciferase experiments and V.E. treated mice by romiplostim and analyzed data. D.M., L.T., and PG contributed to the bone marrow transplantation and mouse study and PR to flow cytometry experiments and analysis. S.N.C., H.R., and J.L.V. designed experiments analyzed the data and wrote the paper.

## Competing interests

The authors declare no competing interests.
