## [Peer Review File · Nature Communications]

Reviewers' comments:

Reviewer #1 (Remarks to the Author):

The authors generated conditional inducible knock-in mouse models for the two most prevalent CALR mutations found in MPN patients, i.e. CALR del52 and ins5. The knock-in mice express a chimeric Calr protein with the exons 1-8 encoded by the mouse Calr gene fused to a hybrid mouse exon 9 with the mutated human C-terminal carrying either the del52 or ins5 mutation. While several mouse models for the CALR del52 mutation have been previously described, the CALR-ins5 knock-in allowed the authors to compare the del52 and ins5 mutations side by side and examine their effects on hematopoiesis.

They found a stronger MPN phenotype in del52 compared to ins5 mutant mice including thrombocytosis, leukocytosis, splenomegaly, bone marrow hypocellularity, megakaryocytic lineage amplification, expansion and competitive advantage of the hematopoietic stem cell compartment. Homozygosity of the CALR mutations amplified these features, e.g. homozygous del52 KI mice display features of a penetrant myelofibrosis-like disorder with extramedullary hematopoiesis linked to splenomegaly, megakaryocyte hyperplasia and the presence of reticulin fibers.

While some differences were noted between the previously published del52 knockin by Li et al. 2018, the basis for these differences have not been worked out by the authors. Overall, the description of the del52 mice does not provide much new information. The novelty is mainly based on the newly generated ins5 knockin and the comparisons of the del52 and ins5 strains. Interesting new information is that both CALR mutations are capable of displacing wild type progenitor and stem cell in competitive transplantation assays and that CALR del52 amplifies the HSC compartment to a higher extent than CALR ins5.

The manuscript could be strengthened by addressing the following issues:

1. To a large part modeling del52 and ins5 mutations in mice successfully recapitulates the differences in phenotypes observed in patients, but some aspects remain discordant, e.g. why MPN patients with ins5 mutations often have higher platelet counts than patients with del52, while the opposite is true in the mouse models. What is the explanation for this difference?
2. Why is the phenotype of heterozygous CALR mutant mice so mild? Most patients with CALR mutations are also heterozygous, but they display a more prominent phenotype, including myelofibrosis.
3. Blood counts: more lineages should be shown in Figure 2. Hemoglobin values would be more informative than showing RBC numbers. Why are lymphocytes and monocyte elevated and not neutrophils? This does not correspond to findings in patients nor to the reported activation of the GCSF-receptor by mutant CALR.
4. What is the genetic background of the mice used genetic background used in the competitive transplantations. The authors mention in the Methods part that 129S2/SvPas ES cell were used.
5. A reference for the different mouse strains used (Flp recombinase, Scl-CreERT , GFP mice etc.) should be given.
6. Why is the amount of the CALR protein in Figure 1c and 1d decreased in the homozygous del52 and ins5 mice compared to heterozygous and wild type controls?

Reviewer #2 (Remarks to the Author):

This is a very timely and important study by Benlabiod et al. Understanding the differences between type I and type II CALR mutations is of great interest in the MPN field, as these mutations have different clinical phenotypes and prognostic outcomes in patients. This is the first study to generate the complement of both type I and type II knock-in mouse models, and to directly compare their phenotypes. These mouse models are an extremely important tool for the study of CALR-mutated MPN.

Major comments:

1. The authors' underlying contention is that the models they have generated are meant to recapitulate the differences in the human phenotypes seen in type I versus type II CALR patients. While this is stated in the Introduction, this idea is not mentioned or revisited until the Discussion, making the manuscript feel simple and descriptive, when in fact the findings are quite important. Added discussion in the results section as to how the phenotypes the authors see in the animals compare to the human phenotype for each mutation type would increase the impact of the results tremendously.
2. Along these lines, why do the authors think they see such a significant increase in T cells and B cells in del52 mice? What is this indicative of? There is no discussion of this finding, and this is not a phenotype seen in human MF patients with CALR mutations.
3. In Figure 1, it would be best to also blot with the mutant CALR antibody to definitively show protein expression of the mutant
4. In Figure 2, it may be better to compare del52 and ins5 heterozygotes on one graph, and del52 and ins5 homozygotes on another graph, rather than comparing hets and homozygous mice within the same genotype. This would be more in line with the crux of the study, which is to compare the phenotypes between the two mutations.
5. In Figure 4, are the authors sure that the H&E spleen panels are 10X? It's very difficult to see the detail (i.e. to discriminate between red and white pulp) at this magnification. Overview pictures of the spleen would help to determine whether the spleen really is fibrotic and to what degree the splenic architecture is disrupted.
6. My biggest concerns with these models are 1) that there is almost no phenotype in the heterozygous mice, which is the closest model of the human disease since CALR mutations are almost always heterozygous, and 2) the degree of splenic fibrosis, which is not typically seen in MF patients. In this case, do these models really faithfully recapitulate the human disease? I worry that the degree of splenic fibrosis may be indicative of an entirely different mechanism/pathway activation unrelated to the established mutant CALR/MPL/JAK/STAT axis activation. A few questions the authors can answer experimentally to allay these concerns:
 - a. Do megakaryocytes from the spleen demonstrate mutant CALR/MPL/JAK/STAT activation? What about from the bone marrow? Some mechanistic studies demonstrating that the phenotype matches the known mutant CALR mechanism would be helpful.
 - b. Is it possible for these mice to get bone marrow fibrosis at all? If the mice are treated with TPO or a TPOR agonist, will they develop BM fibrosis? This would at least demonstrate that there isn't something inherently wrong with the model that is preventing BM fibrosis from developing.
 - c. Are livers in these animals enlarged? Do the livers show fibrosis? Can the authors rule out that the phenotype they see isn't a result of hepatosplenomegaly that has turned fibrotic?
 - d. How long do the mice live once they develop fibrosis? Do the authors have a Kaplan-Meier curve they could show?

Minor comments:

1. In addition to citation 5 (Chachoua et al), the authors should also cite Elf et al (Cancer Discovery 2016) since this is the study that demonstrates physical binding between mutant CALR and MPL, and this is mentioned in the sentence citing Chachoua et al.
2. Page 8 – typo ("commune" instead of "common")

3. Page 10 - typo ("that what" instead of "that was")
4. Page 11 - grammar ("there was barely no difference" should be there was "there was barely a difference")
5. Page 11 - grammar ("the 4-fold increase in BM frequency of giant MKs" should be "the 4-fold increase in the frequency of giant MKs in the bone marrow")
6. Page 11 - grammar ("with no noticeable changes neither in granulo-monocytic progenitors ... nor the ..." should be "with no noticeable changes in the granulo-monocytic ... or the ...")

We would like to thank the reviewers for their time and constructive remarks that helped us improving the manuscript.

Reviewer #1:

1. To a large part modeling *del52* and *ins5* mutations in mice successfully recapitulates the differences in phenotypes observed in patients, but some aspects remain discordant, e.g. why MPN patients with *ins5* mutations often have higher platelet counts than patients with *del52*, while the opposite is true in the mouse models. What is the explanation for this difference?

Indeed, it's a very appropriate comment.

To date, there are only three independent studies from our lab comparing the effect of the two most frequent types of *CALR* mutations in mice including two retroviral mouse models expressing the human *CALR* mutants, and the present work on KI mice that express murine *CALR del52* and *ins5* with the human mutated C-terminal tail chimeric proteins. In these three models, we independently and repeatedly observed that thrombocytosis and the overall phenotype was significantly more severe in presence of *del52* than in presence of *ins5*.

In human ET patients, platelet counts reach slightly but significantly higher levels in presence of *ins5* than *del52*: mean $\times 10^9/L$ (median range) of 1,001 (454-3,460) for *del52* versus 1,199 (520-3,036) for *ins5* ($p=0.03$) in the study from the group of A Tefferi and 832 (502-3,000) for *del52* versus 982 (500-2,670) for *ins5* ($p=0.027$) in the study from Pietra *et al.* While these differences are minor and median intervals suggest that platelet counts can be very heterogeneous between patients, like in KI mice, these differences remain puzzling to us. In both mice and patients, *CALR* mutations induce a continuum of diseases between ET, pre-fibrosis and MF with fluctuating platelet levels. In particular, *del52* ETs are more inclined to progress towards MF than *ins5* ETs suggesting more heterogeneity in platelet counts for *del52* patients. Although allele burden in *CALR*-mutated patients often tends to be close to 50% when disease declare, it would be interesting to compare platelet levels in *del52* and *ins5* ETs with similar and low allele burden and to follow progression of thrombocytosis to measure the real impact of these two mutations on platelet counts (discussed on pages 12-13 of the revised manuscript).

Another of our hypotheses relies on the possible contribution of the homozygous clone in the disease phenotype. Indeed, homozygous clones are more frequent in *ins5* than in *del52* patients. To explore the influence of homozygous clone in the development of the disease we performed a competitive BM transplantation using 20% of homozygous *ins5/ins5* with 80% of heterozygous *+/ins5* KI BM cells (Fig. 8b, c of the revised manuscript). These mice developed a stronger thrombocytosis than mice engrafted with 100% of heterozygous *+/del52* BM cells. Moreover, the homozygous cells had a competitive advantage compared to the heterozygous cells at the HSC level. Thus, in patients it would be interesting to correlate platelet counts with the global *CALR*-mutated allele burden (or percentages of homozygous clones) in MKs (discussed on pages 13-14).

2. Why is the phenotype of heterozygous *CALR* mutant mice so mild? Most patients with *CALR* mutations are also heterozygous, but they display a more prominent phenotype, including myelofibrosis.

We agree with the reviewer that heterozygous *del52*, and to a stronger extent *ins5* KI mice, present an unexpected weak phenotype. In our KI, *CALR del52* is a chimeric protein with the mouse backbone and the human mutated C-terminal tail. In order to understand whether this mild phenotype could be due to a lower than expected activation of murine MPL, we performed a luciferase assay in $\gamma 2A$ cells (Fig. 5a of the revised version). The chimeric *CALR del52* and

ins5 were found to be as efficient than human CALR *del52* and *ins5* in activating murine or human MPL, with STAT5 activation as a read-out. However, both the chimeric and human CALR mutants were twice as less efficient in activating murine MPL compared to human MPL. Moreover, TPO stimulation of CALR *del52*-expressing γ 2A cells (luciferase assay, Fig. 5a) or of homozygous *del52/del52* MKs from KI mice (phosphoflow assay, Supplementary Fig. 7a) induces additional activation and phosphorylation of STAT5 compared to unstimulated condition, respectively, suggesting that murine MPL is not fully activated by the mutants. Finally, we treated wt and homozygous *del52/del52* mice (1 month post-induction of KI expression by tamoxifen) with a TPO agonist (romiplostim) and observed that they developed a significant fibrosis in BM and spleen, on the contrary of untreated mice (Fig. 5b, c, Supplementary Fig. 7b). These data suggest that the phenotype in KI mice is mild probably because constitutive activation of murine MPL by chimeric CALR *del52* and *ins5* is milder compared to the human setting.

3. Blood counts: more lineages should be shown in Figure 2. Hemoglobin values would be more informative than showing RBC numbers. Why are lymphocytes and monocyte elevated and not neutrophils? This does not correspond to findings in patients nor to the reported activation of the G-CSF-receptor by mutant CALR.

We followed the reviewer's advice and added the hemoglobin levels for the heterozygous and homozygous *del52* and *ins5* KI mice as independent panels (e and f in Figure 2 of the revised manuscript). Indeed, hemoglobin values of homozygous *del52* KI mice were significantly lower than control littermates. In addition, as the main goal of the study is to compare the effects of *del52* and *ins5* on hematopoiesis, we re-organized Figure 2 by presenting heterozygous and homozygous values for the two KI on two distinct panels instead of heterozygous and homozygous counts for *del52* KI mice on one side and for *ins5* KI mice on the other side, as suggested by reviewer #2.

We have noticed that data collected from the *del52* KI mice were more heterogeneous than from the wt littermates or even the *ins5* KI mice. While further checking whether CALR *del52* mutant might differently impact various sub-population of T lymphocytes, we also analyzed the levels of circulating neutrophils in 5 additional mice. Adding these values (n=10 for wt and *del52/del52* genotypes in the revised Supplementary Fig. 2b) allowed to detect a significant increase of neutrophils in KI mice compared to wt littermates, as expected from the literature and the described effect of CALR mutants on the G-CSF receptor (lines 104-105 and 310-311). The unexpected leukocytosis observed in KI mice, on the contrary of human ET patients, was explained by a general increase in neutrophils, monocytes, eosinophils, basophils and lymphoid cell populations including several sub-types of T lymphocytes (Supplementary Fig. 2, 3). It is however difficult to know whether this increase is due to CALR immunogenicity and the activation of lymphocytes that remains a matter of debate or, as *CALR* mutations are present in monocytes, B and T lymphocytes, to a direct effect on their expansion. This should be investigated in the future and is discussed on pages 14-15 in the revised Discussion.

4. What is the genetic background of the mice used in the competitive transplantations. The authors mention in the Methods part that 129S2/SvPas ES cell were used.

We thank the reviewer for pointing out this mistake. This has been corrected as "C57BL/6N" on page 18, line 414 of the revised manuscript.

5. A reference for the different mouse strains used (Flp recombinase, Scl-CreERT, GFP mice etc.) should be given.

We added the appropriate references for the different mouse strains: C57BL/6N Tac mice (page 18, line 419); HSC-SCL-Cre-ERT (page 19, line 423) and UBI-GFP/BL6 (page 23, line 540).

6. Why is the amount of the CALR protein in Figure 1c and 1d decreased in the homozygous *del52* and *ins5* mice compared to heterozygous and wild type controls?

We reproduced this Western blot using two different antibody clones from Cell Signaling (now presented on Fig. 1c of the revised manuscript) and Abcam (Supplementary Fig. 1a). The two antibodies (CALR-tot) are directed against the N-terminal portion of the CALR protein that is thus common to CALR both wt and mutants. We repeatedly detected doublet bands at the expected CALR size that might reflect the presence of a cleaved product and/or protein post-translational modifications. We noticed a decrease of CALR in Lin- lysates of homozygous KI mice. After quantification (n=3 different mice/genotype), the 1.8-fold decrease in CALR expression level was significant in the homozygous *del52* context (Supplementary Fig. 1b). This decrease could be due to secretion and/or stability of CALR mutants. Interestingly, using an antibody directed against the mutated C-terminal tail (CALR-Cter), CALR *del52* and CALR *ins5* proteins were barely detectable in heterozygous mice, as opposed to homozygous mice, pointing towards a possible role of CALR wt (present in the heterozygous context) in the secretion and/or the stability of the mutant proteins.

Reviewer #2:

1. The authors' underlying contention is that the models they have generated are meant to recapitulate the differences in the human phenotypes seen in type I versus type II CALR patients. While this is stated in the Introduction, this idea is not mentioned or revisited until the Discussion, making the manuscript feel simple and descriptive, when in fact the findings are quite important. Added discussion in the results section as to how the phenotypes the authors see in the animals compare to the human phenotype for each mutation type would increase the impact of the results tremendously.

We followed the reviewer advice and implemented a few comments in the results section of the revised manuscript to mention when results obtained in KI mice were discrepant with results from patients.

In particular, thrombocytosis and the overall phenotype of KI mice was significantly more severe in presence of *del52* than in presence of *ins5* while, in human ET patients, platelet counts reached slightly but significantly higher levels in presence of *ins5* that *del52*: mean $\times 10^9/L$ (median range) of 1,001 (454-3,460) for *del52* versus 1,199 (520-3,036) for *ins5* (p=0.03) in the study from the group of A Tefferi and 832 (502-3,000) for *del52* versus 982 (500-2,670) for *ins5* (p=0.027) in the study from Pietra *et al.* While these differences are minor and median intervals suggest that platelet counts can be very heterogeneous between patients, like in KI mice, these differences remain puzzling to us. In both mice and patients, *CALR* mutations induce a continuum of diseases between ET, pre-fibrosis and MF with fluctuating platelet levels. In particular, *del52* ETs are more inclined to progress towards MF than *ins5* ETs suggesting more heterogeneity in platelet counts for *del52* patients. Although allele burden in *CALR*-mutated patients often tends to be close to 50% when disease declare, it would be interesting to compare platelet levels in *del52* and *ins5* ETs with similar and low allele burden and to follow progression of thrombocytosis to measure the real impact of these two mutations on platelet counts. We discuss these observations in details in the revised manuscript (pages 12-13).

Another of our hypotheses relies on the possible contribution of the homozygous clone in the disease phenotype. Indeed, homozygous clones are more frequent in *ins5* than in *del52* patients. To explore the influence of homozygous clone in the development of the disease we performed a competitive BM transplantation using 20% of homozygous *ins5/ins5* with 80% of

heterozygous *+/ins5* KI BM cells (Fig. 8b, c of the revised manuscript). These mice developed a stronger thrombocytosis than mice engrafted with 100% of heterozygous *+/del52* BM cells. Moreover, the homozygous cells had a competitive advantage compared to the heterozygous cells at the HSC level. Thus, in patients it would be interesting to correlate platelet counts with the global *CALR*-mutated allele burden (or percentages of homozygous clones) in MKs (discussed on pages 13-14).

2. Along these lines, why do the authors think they see such a significant increase in T cells and B cells in *del52* mice? What is this indicative of? There is no discussion of this finding, and this is not a phenotype seen in human MF patients with *CALR* mutations.

Indeed, the increase in lymphocytes was unexpected (lines 102-103). We further checked whether *CALR del52* mutant might differently impact various sub-population of T lymphocytes (Supplementary Fig. 3 of the revised version). Thus, leukocytosis was due to a general increase in neutrophils, monocytes, eosinophils, basophils and lymphoid cell populations including several sub-types of T lymphocytes (Supplementary Fig. 2, 3). It is however difficult to know whether this increase is due to *CALR* immunogenicity and the activation of lymphocytes that remains a matter of debate or, as *CALR* mutations were demonstrated to be present in monocytes, B and T lymphocytes, to a direct effect on their expansion. This should be investigated in the future and is discussed on pages 14-15 in the revised Discussion.

3. In Figure 1, it would be best to also blot with the mutant *CALR* antibody to definitively show protein expression of the mutant.

We thank the reviewer for this suggestion. We first reproduced the Western blot using two different antibody clones from Cell Signaling (now presented on Fig. 1c of the revised manuscript) and Abcam (Supplementary Fig. 1a). The two antibodies (*CALR-tot*) are directed against the N-terminal portion of the *CALR* protein that is thus common to *CALR* both wt and mutants. We repeatedly detected doublet bands at the expected *CALR* size that might reflect the presence of a cleaved product and/or protein post-translational modifications. We noticed a decrease of *CALR* in Lin- lysates of homozygous KI mice. After quantification (n=3 different mice/genotype), the 1.8-fold decrease in *CALR* expression level was significant in the homozygous *del52* context (Supplementary Fig. 1b). This decrease could be due to secretion and/or stability of *CALR* mutants.

Interestingly, using the antibody directed against the mutated C-terminal tail (*CALR-Cter*), *CALR del52* and *CALR ins5* proteins were barely detectable in heterozygous mice, as opposed to homozygous mice pointing towards a possible role of *CALR* wt (present in the heterozygous context) in the secretion and/or the stability of the mutant proteins (Figure 1c).

4. In Figure 2, it may be better to compare *del52* and *ins5* heterozygotes on one graph, and *del52* and *ins5* homozygotes on another graph, rather than comparing hets and homozygous mice within the same genotype. This would be more in line with the crux of the study, which is to compare the phenotypes between the two mutations.

We agree and followed this advice. As suggested, we re-organized Figure 2 by presenting heterozygous and homozygous values for the two KI on two distinct panels instead of heterozygous and homozygous counts for *del52* KI mice on one side and for *ins5* KI mice on the other side.

5. In Figure 4, are the authors sure that the H&E spleen panels are 10X? It's very difficult to see the detail (i.e. to discriminate between red and white pulp) at this magnification. Overview pictures of the spleen would help to determine whether the spleen really is fibrotic and to what degree the splenic architecture is disrupted.

Indeed, there was a mistake in the magnification of the HES BM panels that was labeled as 40X instead of being 25X. This was corrected. The HES spleen panels are 10X. Overview pictures of the spleen architecture and fibrosis (HES and RET) using a 2.5X magnification were added in Supplementary Figure 6 of the revised manuscript.

6. My biggest concerns with these models are 1) that there is almost no phenotype in the heterozygous mice, which is the closest model of the human disease since CALR mutations are almost always heterozygous, and 2) the degree of splenic fibrosis, which is not typically seen in MF patients. In this case, do these models really faithfully recapitulate the human disease? I worry that the degree of splenic fibrosis may be indicative of an entirely different mechanism/pathway activation unrelated to the established mutant CALR/MPL/JAK/STAT axis activation. A few questions the authors can answer experimentally to allay these concerns: We particularly wish to thank the reviewer for these insightful comments that helped us improving the manuscript.

a. Do megakaryocytes from the spleen demonstrate mutant CALR/MPL/JAK/STAT activation? What about from the bone marrow? Some mechanistic studies demonstrating that the phenotype matches the known mutant CALR mechanism would be helpful.

This was indeed an important point to verify. In order to answer this question, we analyzed MPL activation in BM and spleen MKs measuring STAT5 phosphorylation levels by phosphoflow assay as a readout (Supplementary Fig. 7a). CALR del52 induced the constitutive phosphorylation of STAT5 in homozygous BM MKs compared to wt littermate BM MKs. This phosphorylation was dependent on JAK2 as level returned to basal with the JAK1/2 inhibitor ruxolitinib. Interestingly, *in vitro* stimulation of MPL by TPO induce additional STAT5 phosphorylation, suggesting that 1) some MPL was accessible to TPO when CALR del52 was present and 2) that MPL was only partially activated by the mutant. Moreover, the constitutive activation of STAT5 was comparable in BM and spleen *del52* MKs.

b. Is it possible for these mice to get bone marrow fibrosis at all? If the mice are treated with TPO or a TPOR agonist, will they develop BM fibrosis? This would at least demonstrate that there isn't something inherently wrong with the model that is preventing BM fibrosis from developing.

Following the reviewer's advice, we treated wt and homozygous *del52/del52* mice (1 month post-induction of KI expression by tamoxifen) with a TPO agonist (romiplostim) and observed that they developed a significant fibrosis in BM and spleen accompanied with BM hypocellularity and splenomegaly, on the contrary of untreated mice (Fig. 5b, c, Supplementary Fig. 7b).

In our KI, CALR del52 is a chimeric protein with the mouse backbone and the human mutated C-terminal tail. In order to understand whether this mild phenotype could be due to a lower than expected activation of murine MPL, we performed a luciferase assay in γ 2A cells (Fig. 5a of the revised version). The chimeric CALR del52 and ins5 were found to be as efficient than human CALR del52 and ins5 in activating murine or human MPL, with STAT5 activation as a read-out. However, both the chimeric and human CALR mutants were twice as less efficient in activating murine MPL compared to human MPL. Moreover, TPO stimulation of CALR del52-expressing γ 2A cells also induces additional activation of STAT5 compared to unstimulated condition, suggesting that murine MPL is not fully activated by the mutants.

Taken together, these data suggest that the phenotype in KI mice is mild probably because constitutive activation of murine MPL by chimeric CALR del52 and ins5 is milder compared to the human setting.

c. Are livers in these animals enlarged? Do the livers show fibrosis? Can the authors rule out that the phenotype they see isn't a result of hepatosplenomegaly that has turned fibrotic?

Livers were not enlarged and liver sections that were silver stained did not revealed any significant fibrosis.

d. How long do the mice live once they develop fibrosis? Do the authors have a Kaplan-Meier curve they could show?

From personal observations, we followed mice for as long as two years without noticing any particular differences in the mortality between our KI mice and control littermates. We did not establish a survival curve.

Minor comments:

1. In addition to citation 5 (Chachoua et al), the authors should also cite Elf et al (Cancer Discovery 2016) since this is the study that demonstrates physical binding between mutant CALR and MPL, and this is mentioned in the sentence citing Chachoua et al.

2. Page 8 – typo (“commune” instead of “common”

3. Page 10 - typo (“that what” instead of “that was”)

4. Page 11 – grammar (“there was barely no difference” should be there was “there was barely a difference”)

5. Page 11 – grammar (“the 4-fold increase in BM frequency of giant MKs” should be “the 4 fold increase in the frequency of giant MKs in the bone marrow”)

6. Page 11 – grammar (“with no noticeable changes neither in granulo-monocytic progenitors ... nor the ...” should be “with no noticeable changes in the granulo-monocytic ... or the ...”

We thank the reviewer and have accordingly corrected the revised manuscript.

REVIEWERS' COMMENTS:

Reviewer #1 (Remarks to the Author):

This is an improved revised manuscript. My queries have been adequately addressed.

I have only some minor suggestions:

1. Some improvements in Figure 1 might be helpful for the readers:

Figure 1a: It would be good to indicate in the drawing the position of the homology region used for targeting of the knockin construct.

- Also, it would be nice to show that exon 9 is actually a hybrid composed of a mouse and human sequences (e.g. using a different color within the exon 9 box). Instead of "modified" exon 9 it would be better to call it "mutated exon 9".

- The differences between the wildtype and mutant DNA sequences would be easier to understand if the ins5 mutant was placed below the CALR-wt sequence and the del52 below the ins5 sequence.

Figure 1b: The name of the mouse protein should be "Mouse Calr" (not "Mouse CALR").

- Again I think it would be easier to see the differences between the wildtype and mutant amino acid sequences, if the ins5 mutant was placed below the CALR-wt sequence and the del52 below the ins5 sequence

2. The higher platelet levels in heterozygous CALRdel52 knockin mice published by Li et al 2018 compared with the current model should be discussed. Could it be that using the entire human exon 9 used by Li et al versus using only the human C-terminal frame-shifted sequence of exon 9 makes a difference in the strength of the phenotype? Does the structure-function study by Elf et al Blood 2019 give any clues in that direction? Or is it just the very low levels of the mutant del52 and ins5 protein in the heterozygous mice (Figure 1c)?

3. Do the authors think that the drop in hemoglobin levels after 10 months in del52 and ins5 mice is due to myelofibrosis? Any hemoglobin values determined at time points later than 10 months? Is there a further drop in Hb?

Reviewer #2 (Remarks to the Author):

The authors have performed substantial experiments to address my concern, and the manuscript as a result is much improved.

My primary concerns - 1) whether or not these mice are capable of developing bone marrow fibrosis and not just splenic fibrosis, and 2) whether or not the MPL/JAK/STAT pathway is indeed activated in BM megakaryocytes via mutant CALR - have been sufficiently addressed, and I believe the article is now fit for publication.

We would like to thank the two reviewers for their time and involvement in helping us improving this manuscript. We have addressed the reviewer #1 minor suggestions as followed:

Reviewer #1:

1. Some improvements in Figure 1 might be helpful for the readers:

Figure 1a: It would be good to indicate in the drawing the position of the homology region used for targeting of the knockin construct.

The 5' homology arm (3.4 kb) and 3' homology arm (3.4 kb) encompass a mutated fragment of 4.3 kb including a loxP site, the sequence of the fused exon 8 and exon 9 and a STOP codon, a P_{gk}-NeoR cassette flanked by two FRT sites and the other loxP site followed with exon 8 sequence and the mutated exon 9 (52 bp deletion or 5 bp insertion and humanized sequence with a new STOP codon). We have added the targeting fragment schematic representation in Fig. 1a.

- Also, it would be nice to show that exon 9 is actually a hybrid composed of a mouse and human sequences (e.g. using a different color within the exon 9 box). Instead of "modified" exon 9 it would be better to call it "mutated exon 9".

We followed this advice and modified accordingly Fig. 1a.

- The differences between the wildtype and mutant DNA sequences would be easier to understand if the ins5 mutant was placed below the CALR-wt sequence and the del52 below the ins5 sequence.

We switched the order of the two sequences in Fig. 1a, as recommended.

Figure 1b: The name of the mouse protein should be "Mouse Calr" (not "Mouse CALR").

We have preferred to keep the name of the mouse protein capitalized.

Genes are italicized and, depending on the species of origin, human genes are all in uppercase while mouse and rat genes begin with an uppercase letter followed with the remaining letters in lowercases. Proteins are not italicized and are capitalized depending on species of origin and it has been our understanding that all letters of human and mouse protein symbols are in uppercase.

- Again I think it would be easier to see the differences between the wildtype and mutant amino acid sequences, if the ins5 mutant was placed below the CALR-wt sequence and the del52 below the ins5 sequence.

We switched the order of the two sequences in Fig. 1b, as recommended.

2. The higher platelet levels in heterozygous CALRdel52 knockin mice published by Li et al 2018 compared with the current model should be discussed. Could it be that using the entire human exon 9 used by Li et al versus using only the human C-terminal frame-shifted sequence of exon 9 makes a difference in the strength of the phenotype? Does the structure-function study by Elf et al Blood 2019 give any clues in that direction? Or is it just the very low levels of the mutant del52 and ins5 protein in the heterozygous mice (Figure 1c)?

In the study of Li *et al.*, platelet levels of heterozygous *del52* KI mice reached an average of $3,000 \times 10^3/\mu\text{L}$ as soon as 1 month after induction of the KI expression by poly I:C. Our heterozygous mice developed a thrombocytosis with platelet counts around $2,000 \times 10^3/\mu\text{L}$. The strategies used to generate the two KI models are different: Li *et al.* replaced the totality of murine exon 9 by human mutated exon 9 while we humanized the sequence after the 52-bp

deletion or 5-bp insertion in murine exon 9. However, both strategies result in the same chimera sequence as now shown in the new Supplementary Fig. 10. In this figure we aligned human *CALR del52* exon 9 with our chimeric mouse-human *CALR del52* exon 9 and see that the only mismatches are in the silent position of codons resulting in two identical proteins. Thus, the only difference between the two models is the type of transgenic mice that was crossed with the floxed KI mice, Mx1Cre (ubiquitous expression-can be leaky before induction depending on animal facility sanitary conditions) or SCL-CreER^T (HSC and endothelial cell expression), as we mentioned it (Discussion, page 15). As this point, we do not have another hypothesis to explain the few differences between these two KI models. Moreover, it is almost impossible to compare the level of CALR del52 protein of our heterozygous mice (in Lin⁻ cells) to the protein level measured in spleen cells in Li *et al.* study.

3. Do the authors think that the drop in hemoglobin levels after 10 months in del52 and ins5 mice is due to myelofibrosis? Any hemoglobin values determined at time points later than 10 months? Is there a further drop in Hb?

We do not have further statistical hemoglobin counts for mice older than a year but we assume that the drop in hemoglobin values, that is more variable for the homozygous *ins5* KI mice, could be due to the development of myelofibrosis. The other explanation might be that CALR mutants favor megakaryopoiesis at the expense of erythropoiesis, although there seems to be a partial compensation by the spleen (Supplementary Fig. 5).